# TM4SF19-mediated control of lysosomal activity in macrophages contributes to obesity-induced inflammation and metabolic dysfunction

Cheoljun Choi [1,8], Yujin L. Jeong [2,8], Koung-Min Park [3,8], Minji Kim[1], Sangseob Kim [1], Honghyun Jo [1], Sumin Lee [1], Heeseong Kim[1], Garam Choi [1], Yoon Ha Choi[2], Je Kyung Seong [4], Sik Namgoong [5], Yeonseok Chung [1], Young-Suk Jung [6] ✉, James G. Granneman [7] ✉, Young-Min Hyun [3] ✉, Jong Kyoung Kim [2] ✉ & Yun-Hee Lee [1] ✉

Adipose tissue (AT) adapts to overnutrition in a complex process, wherein specialized immune cells remove and replace dysfunctional and stressed adipocytes with new fat cells. Among immune cells recruited to AT, lipid-associated macrophages (LAMs) have emerged as key players in obesity and in diseases involving lipid stress and inflammation. Here, we show that LAMs selectively express transmembrane 4 L six family member 19 (TM4SF19), a lysosomal protein that represses acidification through its interaction with Vacuolar-ATPase. Inactivation of TM4SF19 elevates lysosomal acidification and accelerates the clearance of dying/dead adipocytes in vitro and in vivo. TM4SF19 deletion reduces the LAM accumulation and increases the proportion of restorative macrophages in AT of male mice fed a high-fat diet. Importantly, male mice lacking TM4SF19 adapt to high-fat feeding through adipocyte hyperplasia, rather than hypertrophy. This adaptation significantly improves local and systemic insulin sensitivity, and energy expenditure, offering a potential avenue to combat obesity-related metabolic dysfunction.

Adipose tissue expansion during overnutrition is associated with the recruitment of specific immune cell subtypes, the functions of which are poorly characterized[1–5]. Mounting evidence indicates that immune cell recruitment is a double-edged sword: on one hand, immune cells support healthy tissue remodeling and expansion; on the other hand, chronic overnutrition often results in persistent recruitment of proinflammatory immune cells that exhibit stalled efferocytosis and contributes to local and systemic insulin resistance. The mechanisms controlling the balance between healthy tissue expansion and persistent inflammation are incompletely understood.

[1]College of Pharmacy and Research Institute of Pharmaceutical Sciences, Seoul National University, Seoul 08826, Republic of Korea. [2]Department of Life Sciences, Pohang University of Science and Technology (POSTECH), Pohang 37673, Republic of Korea. [3]Department of Anatomy and Brain Korea 21 PLUS Project for Medical Science, Yonsei University College of Medicine, Seoul, Republic of Korea. [4]Korea Mouse Phenotyping Center (KMPC), and Laboratory of Developmental Biology and Genomics, College of Veterinary Medicine, Seoul National University, Seoul, Republic of Korea. [5]Department of Plastic Surgery, Korea University College of Medicine, Seoul, Republic of Korea. [6]Department of Pharmacy, College of Pharmacy, Research Institute for Drug Development, Pusan National University, Busan, Republic of Korea. [7]Center for Molecular Medicine and Genetics, Wayne State University, Detroit, MI, USA. [8]These authors contributed equally: Cheoljun Choi, Yujin L. Jeong, Koung-Min Park. ✉e-mail: youngjung@pusan.ac.kr; jgranne@med.wayne.edu; ymhyun@yuhs.ac; blkimjk@postech.ac.kr; yunhee.lee@snu.ac.kr

Macrophage efferocytosis, the process by which macrophages phagocytose and clear apoptotic cells, plays a crucial role in maintaining tissue homeostasis and preventing or resolving inflammation[6,7]. Indeed, recent work demonstrates that interference with macrophage efferocytosis leads to persistent inflammation in several disease models[8,9]. Given the complex nature of adipocyte death, encompassing aspects of various cell death types and mechanisms including necrosis[10], apoptosis[11], and pyroptosis[12], molecular characteristics of the cell types responsible for dead adipocyte clearance remain inadequately understood. However, it is well established that dead or dying adipocytes recruit distinct macrophage subsets characterized by metabolic activation[13], lipid association[4], and lysosomal catabolic activity[14]. Within this context, the role of lipid-associated macrophages that express the triggering receptor expressed on myeloid cells 2 (TREM2) has gained prominence, particularly in diseases marked by lipid stress and inflammation, such as obesity, non-alcoholic steatohepatitis (NASH), and coronary atherosclerosis[4,15,16]. Numerous studies have reported a varied phenotype of these lipid-associated macrophages, including both proinflammatory[17,18] and anti-inflammatory gene expression profiles[5,19], necessitating a more comprehensive characterization.

The lysosomal catabolic activity of lipid-associated macrophages is required for lipid metabolic homeostasis[14] and clearance of lipid remnant from dead adipocytes in hypertrophied adipose tissue[13,14,20]. In particular, vacuolar H + -ATPase (V-ATPase) is critical for lysosomal acidification and catabolic activity. V-ATPase is a large protein complex composed of two multi-subunit domains: peripheral V1 domain involved in ATP hydrolysis and integral (membrane-embedded) V0 domain that transports protons[21,22]. The controlled assembly of the V1-V0 complex represents a key molecular mechanism regulating V-ATPase activity[21].

In obesity, inefficient clearance of dying or dead adipocytes by adipose tissue macrophages leads to the accumulation of dead adipocytes surrounded by TREM2+ lipid-associated macrophages, contributing to insulin resistance and metabolic abnormalities[2,4,17]. We hypothesized that enhancing the efficiency of lipid handling and lysosomal processing could accelerate clearance and resolution. To investigate this, we focused on identifying the factors that are selectively upregulated in TREM2+ macrophages during adipose tissue remodeling induced by high-fat feeding.

Below, we demonstrate that TM4SF19 is a lysosomal protein that is selectively expressed in TREM2+ lipid-associated macrophages that are recruited to adipose tissue during high-fat feeding. Global or macrophage-selective inactivation of TM4SF19 greatly improves systemic metabolism and insulin action in adipose tissues. Mechanistically, we demonstrate that loss of TM4SF19 in macrophages upregulates V-ATPase activities by enhancing V0-V1 assembly and thereby increasing lipid clearance. As TREM2+ macrophages have been implicated in pathophysiological tissue remodeling involving aberrant lipid metabolism, our results indicate that TM4SF19 might be targeted for treating a large class of obesity-associated metabolic diseases.

## Results

### Identification of obesity-induced upregulation of TM4SF19 in adipose tissue macrophages

To identify genes upregulated in obese adipose tissue, we analyzed three publicly available bulk RNA-seq datasets (Fig. 1A, B): gonadal white adipose tissue (GWAT) of normal chow diet (NCD) and high-fat diet (HFD) fed mice (GSE182930)[23], GWAT of wild type and leptin-deficient (ob/ob) mice (GSE150102)[24], and human abdominal subcutaneous WAT of lean subjects and patients with obesity (GSE59034)[25]. Gene ontology (GO) enrichment analysis revealed that the union of the top 100 genes upregulated by obesity from each dataset showed significant enrichment of genes related to the immune system and inflammatory response (Fig. S1A and Supplementary

Table S1). We identified 17 genes upregulated by obesity in all data sets, including genes involved in inflammatory responses [secreted phosphoprotein 1 (Spp1), regulator of G protein signaling 1 (Rgs1), dehydrogenase/reductase 9 (Dhrs9), C-C motif chemokine ligand 2 (Ccl2), macrophage scavenger receptor 1 (Msr1), interleukin 1 receptor antagonist (Il1rn), Cd84] and markers of lipid-associated macrophages (triggering receptor expressed on myeloid cells 2 (Trem2), neutral cholesterol ester hydrolase 1 (Nceh1))[3,4] (Fig. 1B). We focused on Tm4sf19 (GSE182930: 3.64-fold, GSE150102: 6.98-fold, GSE59034: 2.38-fold), a gene with unknown function. TM4SF19 belongs to the transmembrane 4 L six family[26], and the monocyte/macrophage-specific expression of TM4SF19 in human adipose tissue has been previously reported[27–29]. Consistent with the bulk transcriptomic analyses, expression levels of Tm4sf19 mRNA and protein were significantly upregulated in GWAT of HFD-fed mice, while basal expression levels of Tm4sf19 were detectable in bone marrow-derived macrophages (BMDM) (Fig. 1C, D). Furthermore, to determine the cellular distribution of TM4SF19, we isolated F4/80+ cells and floating adipocytes from the GWAT of HFD-fed mice by magnetic activated cell sorting (MACS) and found that Tm4sf19 was highly enriched in F4/80+ cells (Fig. 1E). In addition, publicly available single-cell/nucleus RNA-seq data (GEO: GSE176171)[30] confirmed that TM4SF19 expression is enriched in macrophages in human adipose tissue (Fig. 1F).

Next, we examined the potential upstream regulators of TM4SF19 using the transcription factor binding site prediction tool (Ciiider)[31]. We found that the promoter regions of both mouse Tm4sf19 and human TM4SF19 contain binding sites for nuclear factor-kappa B (NF-κB) and sterol regulatory element binding transcription factor 1 (SREBF1) (Fig. S1B), transcription factors involved in pro-inflammatory responses and cholesterol biosynthesis, respectively. ChIP analysis demonstrated that NF-κB and SREBP1 were recruited to the putative transcription factor binding site of the Tm4sf19 promoter in response to lipopolysaccharide (LPS) and oxLDL treatment, respectively (Figs. 1G, H, and S1C). Protein levels of TM4SF19 and NF-κB (p105 and p50) in RAW 264.7 cells were upregulated by LPS treatment (Fig. 1I), consistent with a previous report[32]. Additionally, oxLDL treatment increased protein levels of TM4SF19 and SREBP1 (including mature forms) (Fig. S1D). TM4SF19 expression levels were positively correlated with body mass index (BMI) and NFKB1 expression in human subcutaneous WAT (Fig. 1J). These data suggest that obesity-induced inflammatory responses can induce Tm4sf19 expression in adipose tissue macrophages (Fig. 1K).

### TM4SF19 is a lysosomal membrane protein that controls V-ATPase activity by interacting with ATP6V0B

To investigate its biological function, we overexpressed TM4SF19 in the RAW264.7 macrophage cell line. We examined the subcellular localization of GFP-tagged TM4SF19 and found that TM4SF19 was specifically localized to the lysosomal membrane, as indicated by LysoTracker (Pearson correlation coefficient = 0.732) and LAMP1 (Pearson correlation coefficient = 0.708) co-staining (Fig. 2A). Next, we focused on lysosomal proteins that potentially interact with TM4SF19. TM4SF19 was found to interact with ATP6V0B in yeast two-hybrid arrays[33]. ATP6V0D2 has been identified as a macrophage-specific component of V-ATPase[34]. Thus, we tested the interaction of TM4SF19 with ATP6V0B and ATP6V0D2, subunits of the membrane-embedded components of V-ATPase. Immunoprecipitation analysis confirmed that TM4SF19 physically interacts with ATP6V0B and ATP6V0D2 (Fig. 2B). Based on the data, we hypothesized that the interaction between TM4SF19 and V-ATPase V0 subunits affects the assembly of V0-V1 domain, which is critical for lysosomal acidification and catabolic activity. To test this hypothesis, the levels of assembled V0-V1 domains (relative abundance of ATP6V1B2 in membrane vs. cytosolic fractions) were quantified in BMDM of WT and TM4SF19 KO mice. Membrane-bound vs. cytosolic ATP6V1B2 levels were increased in

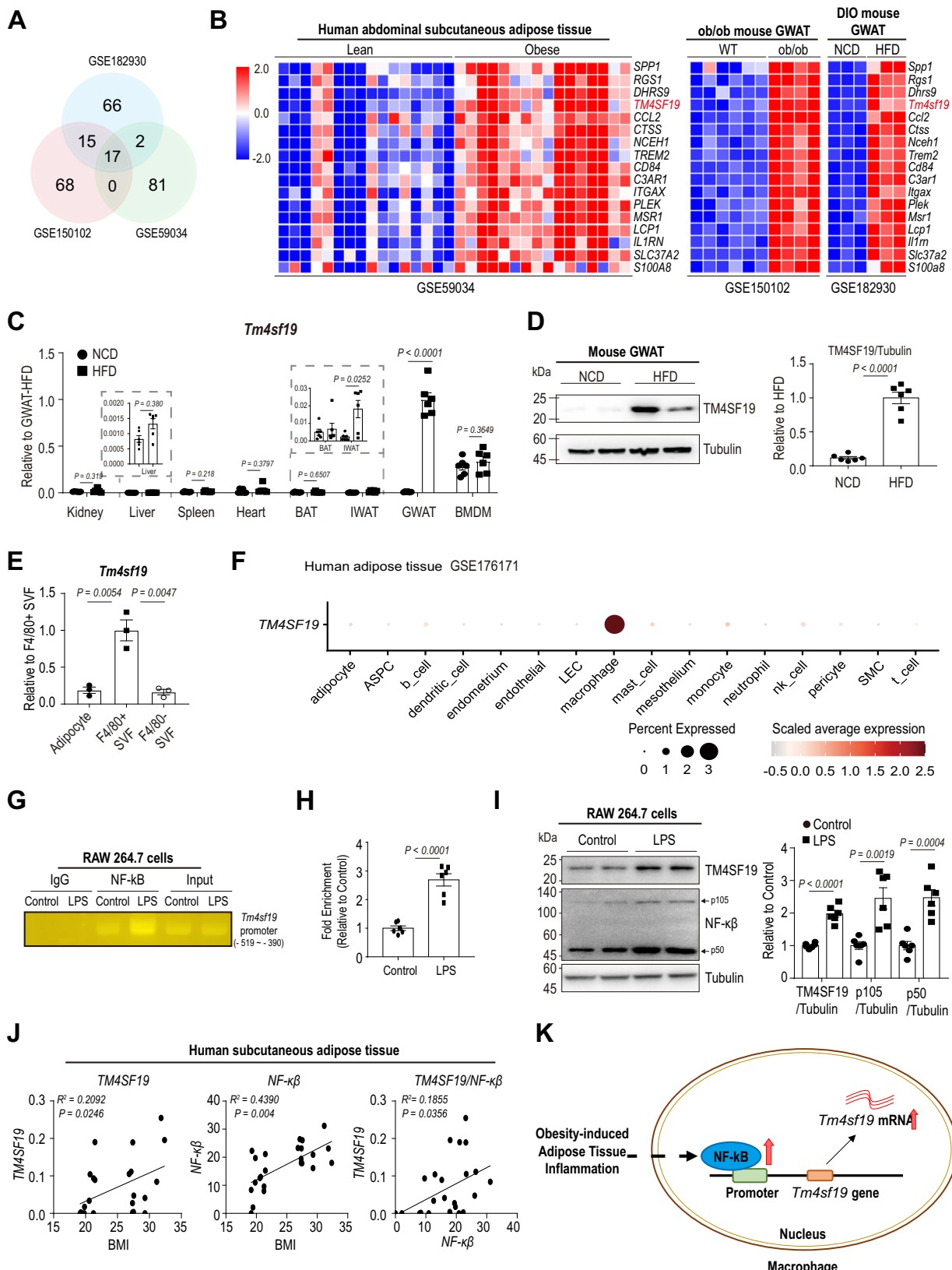

BMDM by TM4SF19 KO, indicating that TM4SF19 KO increases V1-V0 assembly (Fig. 2C). Consistently, TM4SF19 overexpression (OE) increased cytosolic levels of ATP6V1B2, confirming that TM4SF19 inhibits V1-V0 assembly (Fig. 2D).

Because enhanced V1-V0 association is required for V-ATPase activity[35], we measured V-ATPase activity in lysosomal fractions isolated from macrophages (Fig. S2). As expected, TM4SF19 OE decreased lysosomal V-ATPase activity in RAW264.7 cells, while its KO in BMDMs increased activity (Fig. 2E). LysoSensor Yellow/Blue dextran staining confirmed that KO of TM4SF19 in BMDM lowered lysosomal pH (Fig. 2F). Next, we investigated the effects of TM4SF19 KO on the clearance of dying/dead adipocytes by macrophages. TM4SF19 KO

**Fig. 1 | Identification of obesity-induced upregulation of TM4SF19 in adipose tissue macrophages. A** Venn diagrams illustrating overlaps between differentially expressed genes (DEGs: fold change >2, *p*-value < 0.05) from three independent datasets (GSE182930: Transcriptomics profiling of gonadal white adipose tissue (GWAT) of WT mice after normal chow diet (NCD) and high-fat diet (HFD) feeding for 8 weeks, GSE150102: Transcriptomics profiling of GWAT of WT and leptin-deficient mice, GSE59034: Transcriptomics profiling of human subcutaneous WAT of lean and obese subjects). **B** Heatmap of 17 upregulated genes identified in (**A**). **C** *Tm4sf19* mRNA expression levels in various tissues and bone marrow-derived macrophages (BMDMs) of mice after 12 weeks of NCD and HFD feeding (*n* = 6 mice). **D** Immunoblot analysis of TM4SF19 in GWAT of mice after 12 weeks of NCD and HFD feeding (*n* = 6 mice). **E** qPCR analysis of *Tm4sf19* expression in floating adipocytes, and MACS-separated F4/80+ and F4/80- stromal vascular fraction (SVF) isolated from GWAT of HFD-fed mice (*n* = 3 biologically independent replicates). **F.**

*TM4SF19* expression levels in different cell types from human white adipose tissue (GEO: GSE176171). **G, H** Chromatin immunoprecipitation (ChIP) enrichment analysis of recruitment of NF-κB to putative NF-κB binding site of *Tm4sf19* promoter in RAW264.7 macrophages by PCR (**G**) and q-PCR (**H**). PCR amplification was carried out with input and DNA fragments immunoprecipitated by anti-NF-κB antibody and negative control IgG (*n* = 6 biologically independent replicates). **I** TM4SF19 and NF-κB expression levels of RAW264.7 macrophages after LPS treatment for 24 h (*n* = 6 biologically independent replicates). **J** Correlation analysis of *TM4SF19* and *NFKB1* gene expression in human subcutaneous adipose tissue and BMI (*n* = 12 patients). **K** Schematic diagram illustrating the working model of NF-κB-mediated transcriptional control of *Tm4sf19* expression in macrophages during obesity-induced inflammation. Data are presented as mean values ± SEM. *p*-values were determined by the unpaired two-sided Student's *t*-test (**A–E, H, I**), and two-tailed Pearson correlation (**J**). Source Data are provided as a Source Data File.

increased the capacity of BMDM to clear dying/dead C3H10T1/2 adipocytes (Fig. 2G). Furthermore, high magnification images of co-cultures at the early stage (90 mins after co-culture) showed that TM4SF19 KO accelerated phagocytosis as indicated by the number of BMDM-containing intracellular LipidTOX+ lipid droplets (Fig. 2H).

### Single-nucleus RNA sequencing analysis of GWAT dissects cell type-specific effects of *Tm4sf19* deficiency

Next, we investigated the physiological roles of TM4SF19 using a *Tm4sf19* genetic deletion mouse model. *Tm4sf19* KO was confirmed in WAT and BMDM by qPCR analysis (Fig. S3A). TM4SF19 protein was not detectable by immunoblot analysis of adipose tissue of chow diet-fed control mice. As anticipated, in mice fed a normal chow diet, global TM4SF19 KO did not affect body weight, fat tissue mass, adipocyte size, or body composition (Fig. S3B–D). Similarly, TM4SF19 KO did not alter energy expenditure as determined by ANCOVA using body mass as a covariate, either under basal conditions or following treatment with the β3-adrenergic receptor agonist CL316,243 in chow-fed mice (Fig. S3E–G). Furthermore, TM4SF19 KO did not affect food intake, physical activity, and fecal lipid excretion (Fig. S3E and S3H).

We next focused on the effects of *Tm4sf19* deficiency in mice fed a high-fat diet, which greatly upregulates *Tm4sf19* expression. Adipose tissue is a complex and dynamic mixture of immune, stromal, and parenchymal cell types[36]. To investigate cell-type specific effects of *Tm4sf19* deficiency in an unbiased manner, we performed single-nucleus RNA sequencing (snRNA-seq) on whole GWAT from WT and TM4SF19 global KO mice fed with NCD or HFD for 12 weeks (Figs. 3A, B). After filtering out low-quality cells, we profiled the transcriptomes of 54,960 nuclei from two multiplexed replicates for each condition. Cells were clustered using a graph-based clustering algorithm, annotated into 13 cell types based on the expression of curated cell type marker genes for GWAT, and visualized in the uniform manifold approximation and projection (UMAP) representation (Figs. 3C, Fig. S4A, B). Cell-type composition analysis revealed that high-fat feeding expanded the subset of macrophages expressing *Trem2* (Mac.Trem2) and inactivation of *Tm4sf19* partially prevented its expansion (Figs. 3D, E, and S4A). *Tm4sf19* was mainly expressed in Mac.Trem2 subtype as in WT HFD, there is, albeit low, expression in monocytes and lymphatic endothelial cells (Figs. 3F and S4C).

To compare the differentiation trajectories from monocytes to macrophages for each condition, we performed pseudotime analysis for monocytes and macrophage subsets. The constructed trajectory showed three distinct pathways from monocytes to each macrophage subset (Fig. 3G), indicating that the three subtypes have different characteristics. *Lyve1*+ macrophages were designated as perivascular-like macrophages (PVM), which might have a role in angiogenesis and M2-like features, and *Trem2*+ macrophages are known as lipid-associated macrophages (LAM) which expand with HFD feeding[37–39]. Comparing the distribution of cells along the differentiation trajectories, monocytes primarily differentiated into *Lyve1*+ macrophages

(Mac.Lyve1) rather than *Trem2*+ macrophages (Mac.Trem2) with NCD (Figs. 3H, S4D–E). However, with HFD the majority of monocytes underwent differentiation into Mac.Trem2 and TM4SF19 KO attenuated this effect (Figs. 3H, I and S4E), resulting in a decreased Mac.Trem2 population (Figs. 3E and S4F). To examine whether TM4SF19 KO also affects the characteristics of Mac.Trem2, we calculated the signature scores of Mac.Trem2 and Mac.Lyve1 subsets of wild-type mice. TM4SF19 KO decreased the Mac.Trem2 signature score and increased Mac.Lyve1 signature score of Mac.Trem2 in HFD-fed mice, while there was no significant change in NCD-fed mice (Fig. 3J). To confirm this result, we also calculated the signature scores of Mac1 and Mac3 populations, which were considered as *Lyve1*+ and *Trem2*+ macrophages in a previous study[4], respectively, and obtained similar results (Fig. S4G). Consequently, TM4SF19 KO decreases the diet-induced expansion of Mac.Trem2 and alters their expression profile.

### Gene module analysis deconstructs adipose tissue macrophage subpopulations

To further understand the function of *Tm4sf19* in macrophages, we performed co-expression network analysis on monocyte/macrophage populations from WT HFD mice and identified eight co-expressed gene modules (Mod1 ~ 8) (Fig. S5A–C). *Tm4sf19* expression was positively correlated with Mod4 and negatively correlated with Mod2 (Figs. 4A, B and S5B). Genes characterizing Mod4 were enriched in Mac.Trem2, whereas genes in Mod2 were mainly expressed in Mac.Lyve1 (Fig. 4A, B). *Itgax* (*Cd11c*) expression was enriched in *Trem2*+ macrophages (Mod4-enriched), while *Mrc1* (*Cd206*) was abundant in *Lyve1*+ macrophages (Mod2-enriched) (Figs. 4C and S5D). In agreement with our analysis, publicly available single-cell/nucleus RNA sequencing data (GEO: GSE176171)[30] confirmed that *TREM2*+ macrophages are involved in lipid handling and vacuolar acidification, whereas *LYVE1*+ macrophages are involved in phagocytosis and cholesterol efflux in human adipose tissue (Fig. S5E).

Genes that showed a positive correlation with *Tm4sf19* expression (Mod4) were associated with lipid handling markers (*Plin2*, *Pparg*, and *Lipa*), vacuolar acidification (*Atp6v1h*, *Atp6v0d2*, *Atp6v1b2*, *Atp6v0a1*, and *Atp6v1a*), and immune system process (*Cd300a*, *Cd300lb*, *Malt1*, *Alcam*, and *Trim25*), and these gene expression levels were significantly reduced with TM4SF19 KO (Fig. 4C). GO enrichment analysis revealed that molecular functions of V-ATPase were highly enriched in Mod4 gene set macrophage, while cellular components of receptor-mediated endocytosis and complement and coagulation cascades, the mediators of immunity, were associated with genes negatively correlated with *Tm4sf19* expression (Mod2) (Fig. S5F). Genes in Mod2 included anti-inflammatory markers (*Mrc1*, *Clec10a*, and *Rbpj*) and phagocytosis markers (*Mertk*, *Cd163*, *Stab1*) (Fig. 4C). Interestingly, genes that correlated with markers of cholesterol efflux (*Abca1*, and *Apoe*) in adipose tissue were upregulated by TM4SF19 KO (Fig. 4C). Collectively, snRNA-seq analysis refined the distinct roles carried out by obesity-induced macrophage subsets: lipid handing and lysosomal

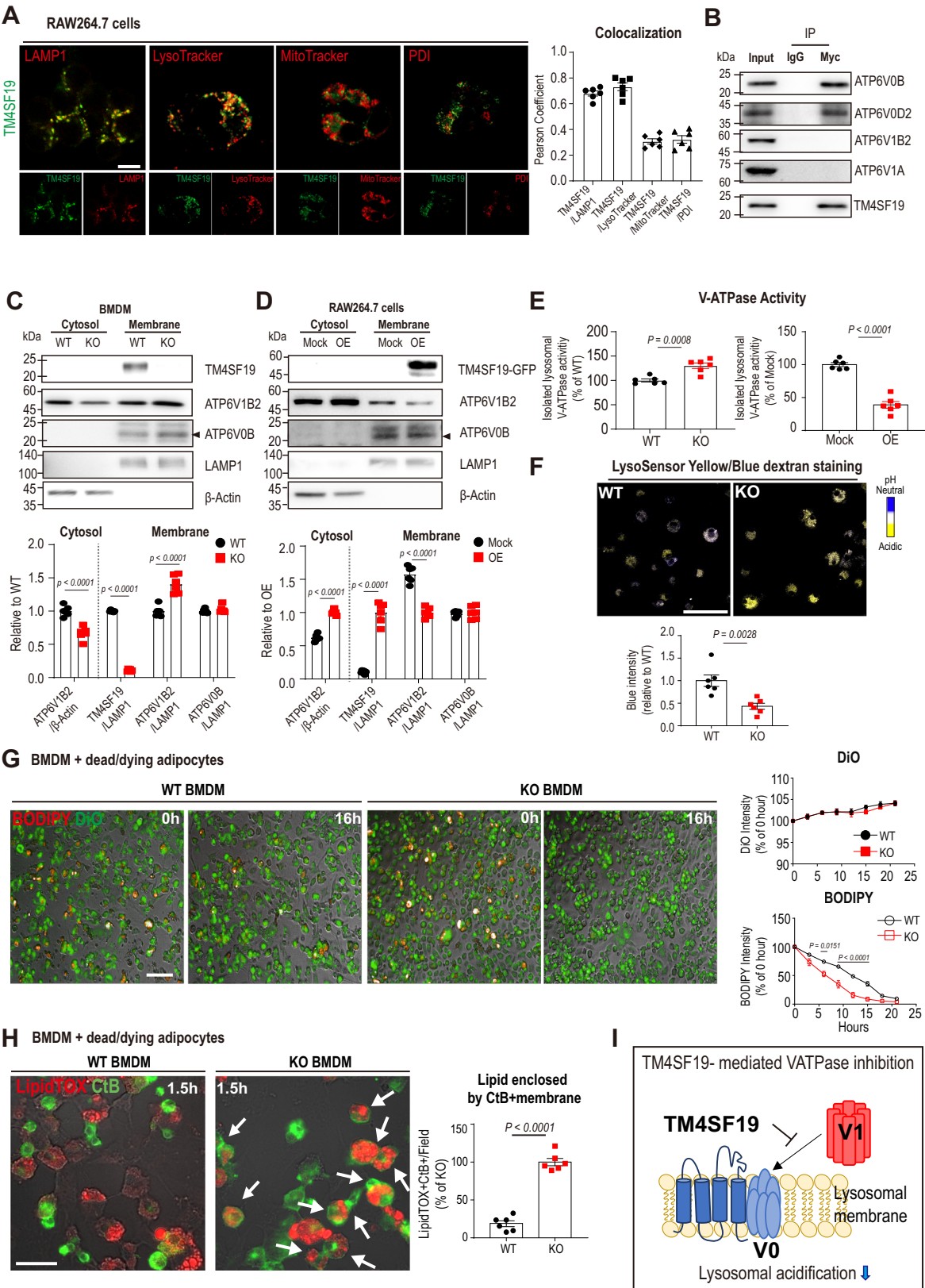

activation, observed in TREM2 + CD11C+ macrophages (Mod4-enriched) and phagocytosis and anti-inflammatory restoration evident in LYVE1 + CD206+ macrophages (Mod2-enriched) (Fig. 4D–G).

To validate our snRNA-seq analysis, we conducted co-culture experiments using RAW264.7 cells and dead/dying adipocytes, monitoring the expression levels of TREM2 and LYVE1 during lipid clearance (Fig. S6A–C). After 8 h of co-culture with dead/dying adipocytes, we observed an upregulation in TREM2 expression, whereas LYVE1 expression decreased (Fig. S6B). However, by the end of the 72-hour co-culture period, the expression levels of *Trem2* and *Lyve1* had undergone a reversal, indicating a gradual transition from a pro-inflammatory *Trem2*+ phenotype into an anti-inflammatory/restorative

**Fig. 2 | TM4SF19, a lysosomal membrane protein, inhibits V-ATPase V1-V0 holoenzyme assembly and lysosomal acidification by interacting with ATP6V0B. A** Representative fluorescence images of RAW264.7 cells overexpressing GFP-tagged TM4SF19 (green) co-stained with the indicated markers (red) (lysosomal associated membrane protein 1(LAMP1), protein disulfide-isomerase (PDI)) (*n* = 6 biologically independent replicates). Scale bar = 5 μm. **B** Immunoprecipitation analysis of Myc-tagged TM4SF19 overexpressing in RAW264.7 cells (representative results from 3 independent experiments). **C, D** Immunoblot analysis of TM4SF19, ATP6V1B2, ATP6V0B, LAMP1, and β-actin protein levels in membrane and cytosolic fractions of WT and TM4SF19 KO BMDM (**C**) (*n* = 6) and TM4SF19 overexpressing RAW264.7 cells (OE) and negative control cells (Mock) (**D**) (*n* = 6 biologically independent replicates). **E** V-ATPase activity of lysosomal fractions from TM4SF19 KO and WT BMDM or TM4SF19 overexpressing RAW264.7 cells and negative controls (*n* = 6 biologically independent replicates). **F** Representative images of BMDM obtained from WT and TM4SF19 KO mice, stained with LysoSensor Yellow/Blue dextran. Yellow fluorescence represents an acidic lysosomal environment, and blue fluorescence represents a neutral lysosomal environment (*n* = 6 biologically independent replicates). Scale bar = 100 μm. **G, H** Long-term live imaging (**G**) and imaging of fixed cells (**H**): WT and TM4SF19 KO BMDMs (stained with DiO: green) cocultured with dying adipocytes (stained with BODIPY: red). Scale bar = 100 μm (**G**). The arrows indicate CtB (green)-stained BMDM that phagocytosed LipidTOX (red)-labeled lipids (Lipid-TOX+CtB+). Scale bar = 20 μm (**H**). The intensity analysis was conducted on each well, with one center field analyzed as six biologically independent replicates (*n* = 6). **I** An illustration of the working model where TM4SF19 interacts with V0 domain of V-ATPase, and inhibits the assembly of V1/V0 complex, resulting in reduced lysosomal acidification. Data are presented as mean values ± SEM. *p*-values were determined by the unpaired two-sided Student's *t*-test (**C**–**H**). Source Data are provided as a Source Data File.

phenotype as macrophage-mediated lipid clearance progresses. Supporting this transition, gene expression levels of *Tnf*, a pro-inflammatory marker, decreased, while those of *Arg1*, an anti-inflammatory marker, increased after 72 h of co-culturing (Fig. S6C).

## TM4SF19 KO prevents HFD-induced inflammation of adipose tissue

Gene set enrichment analysis (GSEA) revealed that TM4SF19 KO downregulated genes involved in the inflammatory response in GWAT and liver of HFD mice, which was confirmed by qPCR (Fig. 5A–C and S7A). In addition, HFD-induced F4/80+ macrophage recruitment in GWAT and liver was significantly decreased in TM4SF19 KO mice (Figs. 5D and S7B). Consistent with qPCR analysis of the macrophage marker, *Adgre1*, flow cytometry analysis (FACS) showed that TM4SF19 KO decreased the proportion of total macrophages (CD45+ CD11B+) in HFD-fed mice (Fig. S7C). As expected from the anti-inflammatory gene signature of adipose tissue of TM4SF19 KO mice, the M2/M1 ratio (M2-like macrophages (CD11B+/CD11C−/CD206+) over M1-like macrophages (CD11B+/CD11C+/CD206−)) was significantly higher in GWAT of TM4SF19 global KO (Fig. 5E). Furthermore, FACS analysis confirmed that TM4SF19 KO reduced TREM2 + CD11C+ macrophages (Mod4-enriched), while enhancing LYVE1 + CD206+ macrophages (Mod2-enriched) in adipose tissue of HFD-fed mice (Fig. 5F, G).

To visualize macrophage dynamics and phagocytosis in vivo, we conducted long-term intravital imaging of adipose tissue using Two-photon microscopy. We employed a *Cx3cr1*-GFP reporter to monitor the motility and morphology of monocytes/macrophages and a *Pdgfra*-Cre/tdTomato reporter to track all progeny of *Pdgfra* expressing cells (including adipocytes/adipocyte progenitors) in both TM4SF19 KO and WT genetic backgrounds, respectively. After subjecting the mice to 8 weeks of high-fat diet feeding, we observed that in WT reporter mice, GFP-positive cells were predominantly found in clusters surrounding dead adipocytes, resulting in the formation of crown-like structures (CLS) (Figs. 5H and S7E, F). Two-photon intravital imaging and H&E staining of adipose tissue further confirmed that TM4SF19 KO reduced the frequency of crown-like structures (Fig. S7E, F). In contrast, macrophages of TM4SF19 KO mice migrated faster and further than WT, which were relatively static (Fig. 5H, I and Supplementary Video 1). Furthermore, representative migration plots showed increased levels of velocity and displacement of macrophages in TM4SF19 KO mice (Fig. 5I). Remarkably, we were able to observe vigorous phagocytosis of adipocytes by macrophages in TM4SF19 KO mice (Fig. 5J and Supplementary Video 2). This in vivo visualization further supports the notion that the resolution and clearance of dying/dead adipocytes by macrophages increased in TM4SF19 KO mice.

## TM4SF19 KO restores insulin-sensitive adipocytes in adipose tissue of mice fed HFD

Adipose tissue adapts to overnutrition by adipocyte hypertrophy and hyperplasia. However, adipocyte hypertrophy is a hallmark of dysfunctional adipose tissue and is associated with insulin resistance and metabolic dysfunction[40,41]. We noted that TM4SF19 KO decrease the size of GWAT adipocytes without a proportionate reduction in fat mass, suggesting adipocyte hyperplasia (Fig. 6A). To quantitatively assess this phenomenon, we employed the method described by the Gallian group to estimate the number of adipocytes in the WAT[42]. Notably, TM4SF19 KO increased adipocyte number in GWAT compared to WT mice (Fig. 6B). To test whether TM4SF19 KO increases de novo adipogenesis from proliferating preadipocytes, we utilized labeling and tracing with thymidine analogue BrdU (Fig. 6C) and found that TM4SF19 KO increased the numbers of BrdU+PLIN1+ adipocytes in GWAT (Fig. 6D).

We then used our snRNA-seq data to further characterize adipocytes and adipocyte progenitor cells (APCs), which are the major cell types that respond to nutritional stimuli in adipose tissue. Sub-clustering analysis identified two populations of APCs (*Pdgfra, Dcn*, and *Col1a1*) and three populations of mature adipocytes (*Adipoq, Cidec, Pparg, Plin4*, and *Lep*) (Fig. 6E, F). Three adipocyte populations were characterized by graded expression of adipocyte marker genes. *Adipoq* was expressed in Adipocyte.1, and *Plin4* and *Lep* were highly expressed in Adipocyte.3 compared to Adipocyte.1. Adipocyte.2 showed intermediate expression of these genes (Figs. 6F and S8A). APCs could be subdivided into APC.Dpp4 (*Dpp4, Pi16*, and *Cd55*) and APC.Icam1 (*Icam1, Col4a1*, and *F3*), characterized by the known APC subtype marker genes (Fig. 6F). Pseudotime analysis of adipocytes and APCs revealed the differentiation trajectories from APCs to adipocytes, divided into two pathways, enriched in adipocytes of WT NCD-fed and WT HFD-fed conditions, respectively (Figs. 6E, F and S8B). In the WT NCD-fed condition, Adipocyte.1 occupied the largest population, with 73% of the total adipocytes and Adipocyte.3 was rarely detected (less than 3%). In contrast, in WT HFD-fed mice, Adipocyte.3 represented approximately 59% of the total adipocytes and became the major population among adipocytes (Fig. 6H). Interestingly, TM4SF19 KO partially restored the Adipocyte.1 population in HFD-fed mice (Odds ratio = 3.71, *p* = 0.0002) whereas there was no difference in NCD-fed mice (Fig. 6G, H). This observation suggests that TM4SF19 KO potentially increases de novo adipogenesis from pro-genitors, generating new adipocytes with improved metabolic functions.

To characterize the difference between Adipocyte.1 and Adipocyte.3, we performed GO enrichment analysis on differentially expressed genes (DEGs) between Adipocyte.1 and Adipocyte.3. This analysis revealed that genes involved in insulin response and lipolysis were upregulated in Adipocyte.1, while genes associated with lipid biosynthesis and apoptotic process were enriched in Adipocyte.3 (Fig. 6I). Genes up-regulated in Adipocyte.1 included adipokines such as *Adipoq* and *Cfd*, which have beneficial effects on insulin sensitivity[43,44] and expression of insulin signaling genes (Fig. 6J). Adipocyte.1 showed increased lipolysis and decreased lipid accumulation (Fig. 6J). In contrast, Adipocyte.3 showed the opposite gene expression

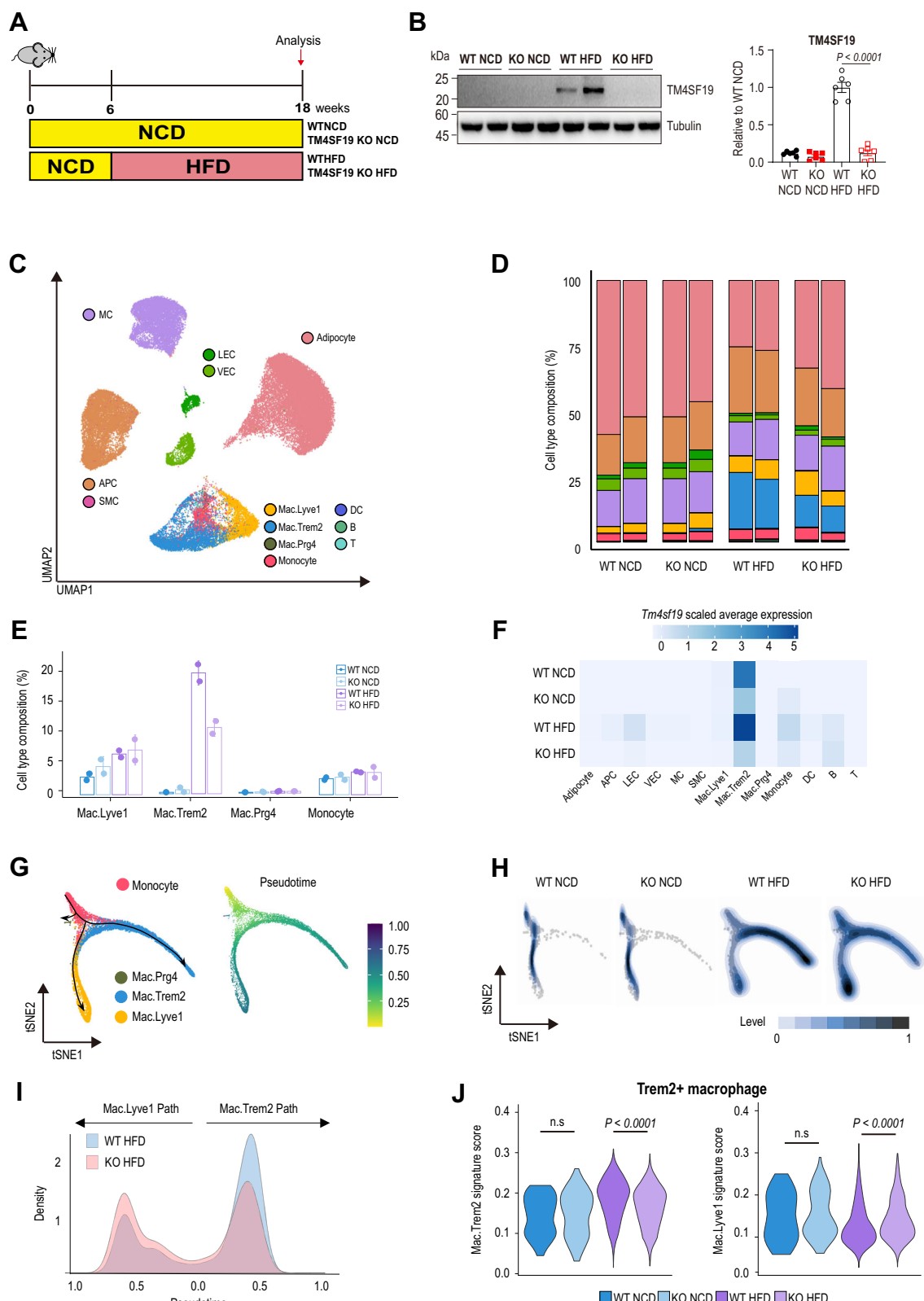

patterns compared to Adipocyte.1, including decreased insulin sig-naling and lipolysis, but increased lipid accumulation (Fig. 6J). These results indicate that the Adipocyte.1 population, which was the major population in NCD-fed mice adipocytes, has transcriptional signatures associated with insulin sensitivity, and the expansion of Adipocyte.1 in TM4SF19 KO mice contribute to restoration of adipose tissue insulin sensitivity during high-fat feeding.

Furthermore, we confirmed that Adipocyte.1-specific markers (*IRS1, INSR*, and *PDE3B*) in human adipose tissue tended to be nega-tively correlated with BMI, whereas Adipocyte.3-specific markers (*LEP* and *FASN*) were positively correlated with BMI (Fig. 6K), as previously reported[30]. In addition, we found that *PDE4D*, which affects the development of insulin resistance in obesity[45,46], was positively corre-lated with BMI (Fig. 6K).

**Fig. 3 | *Tm4sf19* is expressed in *Trem2*+macrophages and TM4SF19 KO reduces HFD-induced recruitment of *Trem2*+ macrophages in GWAT. A** Experimental strategy of snRNA-seq analysis of HFD-fed TM4SF19 KO mice. **B** Confirmation of TM4SF19 expression levels in GWAT of NCD or HFD-fed TM4SF19 KO and WT mice by immunoblot analysis (*n* = 6 mice). **C** UMAP plot of total 54,960 nuclei isolated from GWAT of WT and TM4SF19 KO mice fed NCD or HFD for 12 weeks. (two replicates per condition). Clusters are colored by cell types: Adipocyte, mesothelial cell (MC), lymphatic endothelial cell (LEC), vascular endothelial cell (VEC), adipocyte progenitor cell (APC), smooth muscle cell (SMC), *Lyve1*+macrophage (Mac.*Lyve1*), *Trem2*+macrophage (Mac.*Trem2*), *Prg4*+macrophage (Mac.*Prg4*), monocyte, dendritic cell (DC), B cell and T cell. **D** Total cell type composition of NCD and HFD-fed WT and TM4SF19 KO mice in each sample. **E** Monocyte and macrophage subtype composition in total cell population for each condition (WT NCD, TM4SF19 KO NCD, WT HFD, TM4SF19 KO HFD). The error bar represents mean ± SD (*n* = 2 mice). **F** Average *Tm4sf19* expression levels of each cell type in WT and TM4SF19 KO mice fed NCD and HFD. **G** tSNE plots showing in silico pseudotime analysis of monocyte and macrophage subtypes. Cells were colored by cell types (left) and pseudotime (right). The arrows on the left panel show differentiation directionality. **H** tSNE plots split by NCD or HFD condition. The color indicates the density of cells. **I** Density plot showing the distribution of cells into *Lyve1*+macrophage (Mac.*Lyve1*) path and *Trem2*+macrophage (Mac.*Trem2*) path of HFD-fed WT and TM4SF19 KO mice according to pseudotime. **J** Mac.*Lyve1* (left) and Mac.*Trem2* (right) gene signature scores in *Trem2*+macrophage in each condition (WT NCD, TM4SF19 KO NCD, WT HFD, TM4SF19 KO HFD). ns: *p* > 0.05. (*p*-values for mac.*Trem2* signature score = 2.03e-36, mac.*Lyve1* signature score = 5.98e-22). Data are presented as mean values ± SEM (**B**) or SD (**E**). *p*-values were determined by the unpaired two-sided Student's *t*-test (**J**) and two-way ANOVA followed by Bonferroni post hoc tests (**B**). Source Data are provided as a Source Data File.

## TM4SF19 KO prevents HFD-induced obesity and insulin resistance of adipose tissue

We next examined the effect of TM4SF19 KO on metabolic function during high-fat feeding. Global TM4SF19 KO reduced the weight gain of mice over the 12 weeks of HFD feeding (Fig. S9A–C). Furthermore, glucose and insulin tolerance were improved in HFD-fed TM4SF19 KO mice (Fig. S9D–F). Global TM4SF19 KO also corrected HFD-induced hyperinsulinemia and hyperlipidemia (TG and cholesterol) (Fig. S9G, H). As expected from snRNA-seq data (Fig. 6), TM4SF19 KO greatly improved in vivo insulin action as assessed by insulin-dependent phosphorylation of AKT and IRS1/2 (Fig. S9I).

Furthermore, HFD-fed TM4SF19 KO mice demonstrated higher energy expenditure compared to WT controls, even after adjusting for body mass using ANCOVA[47], without affecting food intake, physical activity, and fecal fat excretion (Figs. 7A, B and S10). Consistent with elevated energy expenditure, GSEA analysis indicated that TM4SF19 KO prevented HFD-induced reduction in mitochondrial biogenesis and oxidative phosphorylation gene expression (Fig. 7C, D), which was confirmed by western blot analysis (Fig. 7E).

## Macrophage-specific TM4SF19 KO protects obesity-related metabolic dysfunction

Finally, we established a tamoxifen-inducible macrophage-specific TM4SF19 KO mouse model (TM4SF19 MKO) by crossing *Tm4sf19*^flox/flox mice with *Csf1r*-CreER mice. After tamoxifen induction, macrophage-specific *Tm4sf19* deletion was confirmed in GWAT of TM4SF19 MKO mice (Fig. 8A). The HFD-induced upregulation of TM4SF19 expression was not detectable in GWAT of TM4SF19 MKO mice (Fig. 8B). Consistent with global TM4SF19 KO mice, HFD-induced increases in body weight, adipose tissue mass, and body fat mass were suppressed by TM4SF19 MKO (Fig. 8C–E). TM4SF19 MKO also improved glucose tolerance and insulin sensitivity (Fig. 8F, G). Indirect calorimetry analysis at 5 weeks and 12 weeks of HFD-feeding indicated that TM4SF19 MKO enhanced oxygen consumption and energy expenditure in HFD-fed mice, independent of changes in body weight. This occurred without affecting locomotor activity, food intake, and fecal fat excretion (Figs. 8H, I, and S11). Furthermore, TM4SF19 MKO decreased the expression of genes associated with pro-inflammatory responses [C-C motif chemokine ligand 2 *(Ccl2)*] and F4/80+ macrophage markers *(Adgre1)* during HFD feeding (Fig. 8J, K). Consistent with TM4SF19 global KO, macrophage-specific TM4SF19 KO reduced LAM in GWAT and decreased the proportion of LAM (TREM2+F4/80+CD45+) in the stromal vascular fraction (SVF) isolated from GWAT (Fig. 8L, M). Furthermore, TM4SF19 MKO prevented the recruitment of total macrophages (CD45+F4/80+) by HFD feeding (Fig. 8L). The lipid accumulation in macrophages (CD45+F4/80+BODIPY+) and total CD45+ cells (CD45+BODIPY+) were also reduced by TM4SF19 MKO (Fig. 8M).

As an alternative approach, we performed bone marrow transplantation (BMT) of TM4SF19 KO immune cells to WT mice to selectively replace TM4SF19 in macrophages and fed an HFD for 12 weeks (Fig. S12A). We confirmed that TM4SF19 was successfully removed (>90%) from GWAT of BMT KO HFD mice after irradiation (Fig. S12B). Consistent with HFD-fed TM4SF19 KO mice, BMT KO mice gained less weight on HFD feeding than BMT WT mice (Fig. S12C). BMT KO significantly diminished the weights of WAT in HFD-fed mice compared to BMT WT mice (Fig. S12D). Fasting glucose and insulin levels, as well as glucose and insulin tolerance, were improved in HFD-fed BMT KO mice (Fig. S12E–H). Consistent with the snRNA-seq analysis (Fig. 3E), the proportion of TREM2+ macrophages (CD45+TREM2+F4/80+) (Fig. S12J) and LAMs (CD45+F4/80+BODIPY+) were reduced in GWAT of HFD-fed BMT KO mice (Fig. S12K).

## Discussion

Overnutrition induces a persistent inflammatory state in adipose tissue, leading to systemic metabolic dysfunction, yet the precise mechanisms governing the balance between inflammation and resolution remain poorly understood. TREM2+ lipid-associated macrophages are recruited to adipose tissue during high fat feeding where they are reported to be involved in the clearance of lipid-laden apoptotic cells[9,48] and tissue remodeling[4]. Despite its importance in tissue remodeling, macrophage phagocytosis can become stalled, resulting in the accumulation of CLS surrounding dead adipocytes[2] and unresolved inflammation[5]. The factors that control the efficiency of this phagocytic process are not well understood. Thus, we hypothesized that factors improving macrophage phagocytosis would facilitate tissue restoration and enhance systemic metabolism. Based on this premise, we focused on the identification of genes enriched in TREM2+ macrophages and differentially regulated by HFD.

This study demonstrated that TM4SF19, formerly a protein of unknown function, is upregulated in TREM2+ macrophages that are recruited to adipose tissue during high-fat feeding. We discovered that TM4SF19 is a macrophage-enriched lysosomal membrane protein that represses lysosomal V-ATPase activity by interacting with V-ATPase V0 domain. Based on this mechanism, we postulated that TM4SF19 acts as a brake, limiting the efficient clearance of lipid-stressed adipocytes. As anticipated, TM4SF19 knockout expedites the clearance of dead fat cells, protecting against obesity-induced inflammation and metabolic dysfunction. These observations shed new light on the mechanisms by which specialized macrophage subsets regulate the balance between inflammation and restoration.

We demonstrated that TM4SF19 interacts with V-ATPase V0 domain in macrophages, interfering with V1/V0 association. Interestingly, recent work has found that V-ATPase regulates the biosynthesis of pro-resolving lipid mediators in M2-like macrophages[49], whereas inhibition of lysosomal acidification promotes proinflammatory cytokine secretion from M1-like macrophages[50] and induces macrophage-proinflammatory activation[51]. Further investigation is required to fully understand the multiple molecular mechanisms of *Tm4sf19* KO-

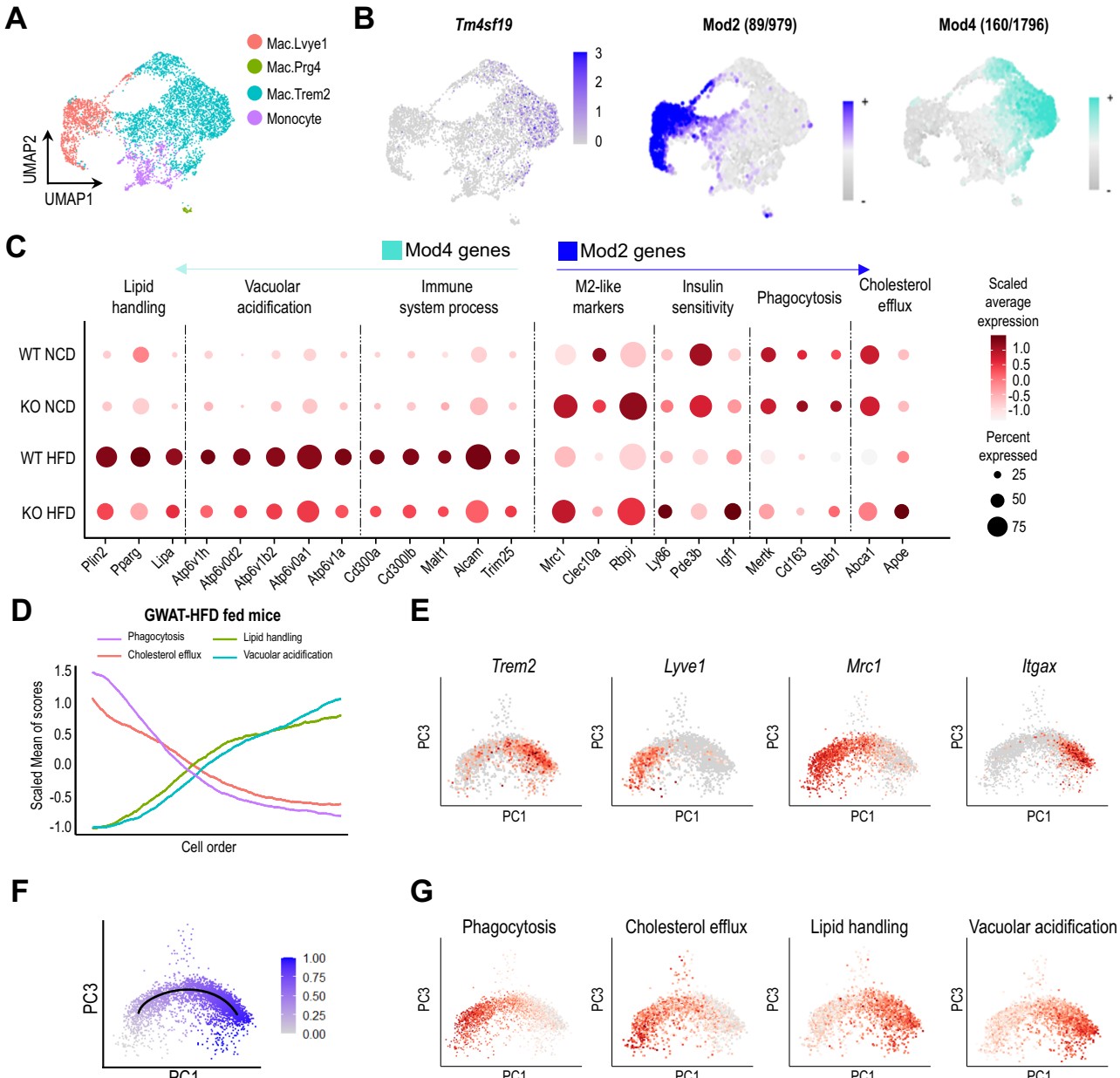

**Fig. 4 | Gene module analysis deconstructs adipose tissue macrophage subtypes. A** UMAP plot of monocyte and macrophage population from WT HFD condition. Four different colors demonstrate the different monocyte/macrophage cell types. **B** *Tm4sf19* gene expression level (left) and module eigengenes (MEs) score of Mod4 and Mod2 (right). *Tm4sf19* gene expression in the population of monocyte/macrophage from WT HFD is included in Mod4. The number of genes in each module and down- (Mod4) or up-regulated (Mod2) in the TM4SF19 KO HFD condition relative to the WT HFD condition is indicated. **C** Gene expression in monocyte and macrophage subtypes from WT NCD, KO NCD, WT HFD, and KO

HFD condition associated with biological terms chosen from Mod4 down-regulated genes and Mod2 up-regulated genes. **D** Graph showing the module scores for each biological term in cells ordered along the trajectory from *Trem2*- to *Trem2*+ macrophages of GWAT of HFD-fed mice. **E** Expression levels of *Trem2, Lyve1, Mrc1,* and *Itgax* in macrophages. **F** Trajectory from *Trem2*- to *Trem2*+ macrophages on the PCA plot. The cell order along the trajectory is indicated by color. The PCs were selected to distinctly separate the *Trem2*+ and *Trem2*- macrophages. **G** Module scores for each biological term in the macrophage population are shown in a PCA plot. Source data are provided as a Source Data file.

mediated beneficial effects on obesity-induced inflammation and metabolic dysfunction.

Our snRNA-seq analysis demonstrates that *Tm4sf19* KO reverses insulin resistance in adipocytes. Because TM4SF19 is specifically expressed in macrophages, the direct KO effect was limited to these cells. Thus, the observed effects on systemic metabolism likely involve indirect pathways, such as reduced inflammation. We further validated that *Tm4sf19* KO-mediated facilitation of dead adipocyte clearance leads to a reduction in the steady-state levels of crown-like structures and promoted the formation of new fat cells with improved metabolic

function. Prior research has demonstrated that proinflammatory macrophages stall de novo adipogenesis and inhibit mitochondrial biogenesis[52–54]. Consequently, the enhanced phagocytic activity of macrophages by TM4SF19 KO facilitates restorative adipose tissue remodeling, leading to increased systemic energy expenditure and improved insulin sensitivity. While we considered *Mertk*[55], *Cd163*[56], and *Stab1*[57] as potential phagocytosis markers in our snRNA-seq data analysis, further studies are needed to conclusively establish the phagocytic activity of macrophage subtypes identified in the current study.

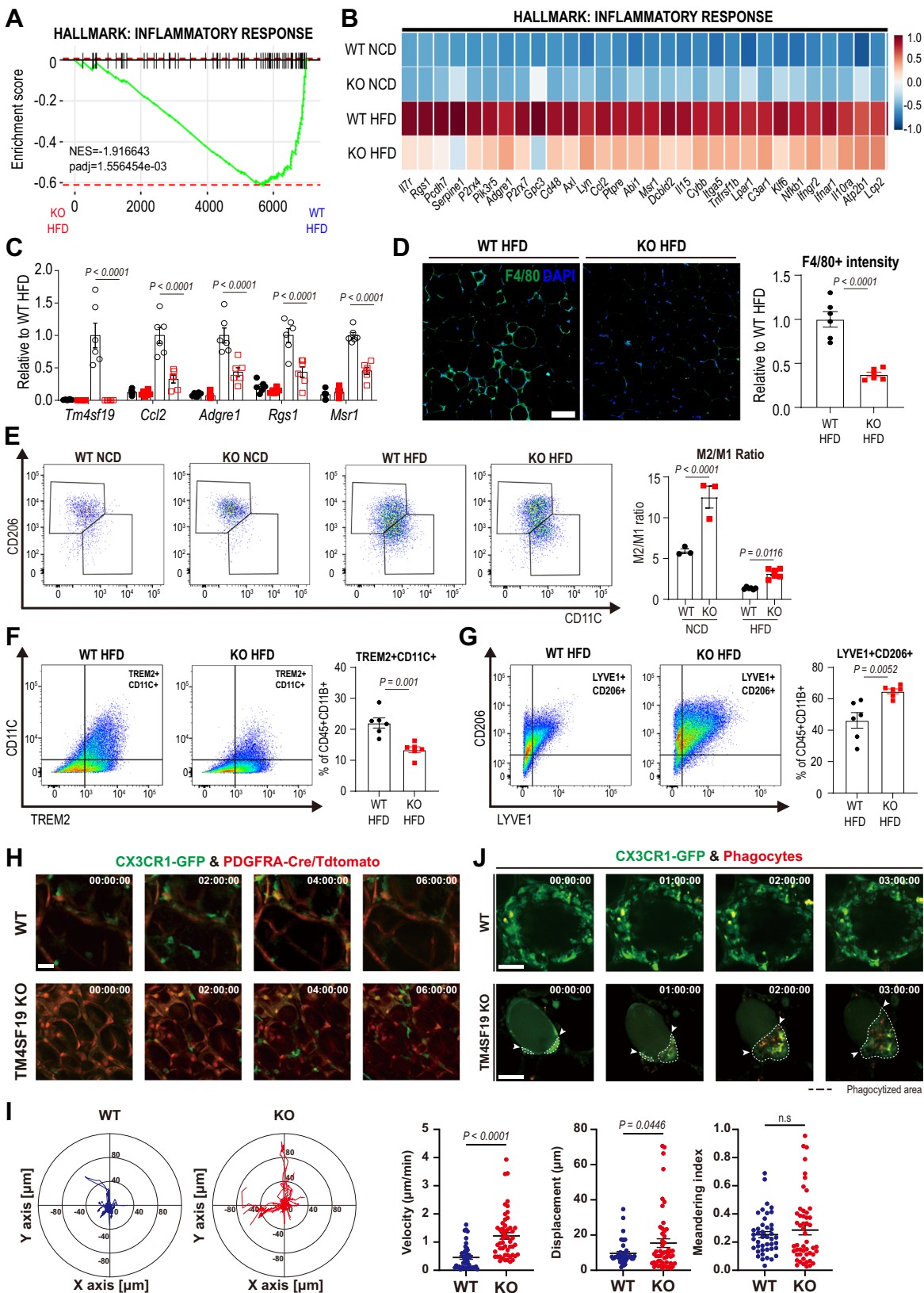

TREM2+ macrophages are recruited to various tissues during overnutrition, and are thought to be involved in the pathogenesis of diseases, such as NASH[9,58] and atherosclerosis[59]. Our findings provide a foundation for investigating the contribution of TM4SF19 in these and other disease contexts. Furthermore, with regard to the functions of TM4SF19, interventions targeting mechanisms that improve lysosomal functions, including lipid handling and phagocytotic capacity of macrophages, could potentially lead to metabolic improvement and hold significant therapeutic implications.

In conclusion, our study demonstrates that TM4SF19 deletion in macrophages protects mice from obesity-induced metabolic dysfunction and suggests that TM4SF19-mediated control of lysosomal

**Fig. 5 | TM4SF19 deletion reduces HFD-induced inflammatory responses of adipose tissue. A** GSEA showing enrichment in inflammatory response (NES = −1.916643, adjusted *p*-value = 1.556454e-03) by genes significantly contributed to all cells of GWAT of WT HFD and TM4SF19 KO HFD mice. Among the variable genes ordered according to WT HFD and TM4SF19 KO HFD group loading, genes associated with the inflammatory response were marked with a black vertical line. **B** Heatmap of the downregulated genes involved in the inflammatory response in all cells of TM4SF19 KO HFD compared to WT HFD mice. Expression levels were scaled across conditions. **C** mRNA expression levels of genes involved in the inflammatory response in GWAT of each condition (*n* = 6 mice). **D** Immunofluorescence staining of F4/80 (green) with DAPI (blue) counterstaining in paraffin sections. Scale bar = 100 μm. **E−G** Representative flow profiles of CD206 (M2) and CD11C (M1) (**E**), TREM2 and CD11C (**F**), and LYVE1 and CD206 (**G**) expression levels in cells from GWAT of WT and TM4SF19 KO mice fed NCD (*n* = 3 mice) or HFD (*n* = 6 mice) for 12 weeks. **H** Time lapse images from two-photon intravital imaging of CX3CR1-GFP+ macrophages (green) in WT and TM4SF19 KO mice. PDGFRA-tdTomato-reporter (red) expression visualized PDGFRA+ cells and PDGFRA+ progenitor-derived adipocytes. Scale bars = 30 μm. (three biologically independent animals per each condition, showing representative images from a total of six fields: two fields/mouse). **I** Representative migration plots in the 2-dimensionally projected graph from 3-dimensional intravital images, showing macrophage migration trajectories in WT (*n* = 43 GFP+ cells from six fields, three biologically independent animals) and TM4SF19 KO (*n* = 52 GFP+ cells from six fields, three biologically independent animals) PDGFRA-Cre/LSL-tdTomato/CX3CR1-GFP mice (Also See Supplementary Movie 1). **J** Two-photon intravital imaging of an adipocyte phagocytized by macrophages in adipose tissue. CX3CR1-GFP signals (green) and WGA-blue (red in Figure) visualized phagocytes. The images in the lower panel indicated the destructed area of an adipocyte with dashed lines and the front line of the phagocytic progress with the arrows. Scale bars = 30 μm. (Also See Supplementary Movie 2). Data are presented as mean values ± SEM. *p*-values were determined by the unpaired two-sided Student's *t*-test (**D**, **F**, **G**, **I**) and two-way ANOVA followed by Bonferroni post hoc tests (**C**, **E**). Source data are provided as a Source Data file.

activity is a potential therapeutic target to facilitate the resolution of obesity-induced adipose tissue inflammation and related disorders.

## Methods

All research conducted for this manuscript complies with ethical regulations. All animal protocols were approved by the Institutional Animal Care and Use Committee at Seoul National University (SNU-211209-4-1, SNU-230130-5, SNU-210403-2) and conducted in strict compliance with the guidelines for humane care and use of laboratory animals specified by the Ministry of Food and Drug Safety. Human studies were conducted in accordance with the Declaration of Helsinki. All patients provided written informed consent, and ethical approval was obtained from the Korea University Guro Hospital (KUGH) Ethics Committee (Institutional Review Board No. 2022GR0095).

### Mice

All mice used in the experiments for this study were male, aged between 6 and 16 weeks old. Mice were housed at 22 ± 1 °C and maintained on a 12-h light/12-h dark cycle with free access to food and water. Mice were euthanized by cervical dislocation. C57BL/6N-*Tm4sf19*^em1cyagen (TM4SF19 KO; Cyagen, KOCMP-20925-Tm4sf19) mice were established by the CRISPR/Cas9 system with the use of single guide RNA (5′-CTGACTCGTCTAGTGG-3′). The primers used for genotyping to confirm TM4SF19 KO were as follows: forward: 5′-TTA-GAGGAAGTCCTTGAGACCCC-3′ and Reverse: 5′- ATCTGTATGACC GTTCACTTTGGA-3′.

To introduce *Cx3cr1*-GFP reporters for macrophages and monocytes and *Pdgfra*-Cre_tdTomato reporters for all progeny of PDGFRα + cells (adipocyte progenitors and adipocytes), we employed the following transgenic mouse strains: B6.Cg-Gt(ROSA) 26Sor^tm9(CAG-tdTomato)Hze/J (Jackson laboratory, stock# 007909), C57BL/6-Tg(*Pdgfra*-cre)1Clc/J (Jackson laboratory, stock# 013148), B6.129P2(Cg)-*Cx3cr1*^tm1Litt/J (Jackson laboratory, stock# 005582). We established reporter mice with global knockout of TM4SF19 or WT for intravital imaging experiments.

Inducible macrophage-specific TM4SF19 KO (*Tm4sf19*^flox/flox/*Csf1r*-CreER: TM4SF19 MKO) mice were generated by crossing *Tm4sf19*^flox/flox mice (C57BL/6JSmoc-*Tm4sf19*^em1(flox)Smoc, Shanghai model organism, stock#: NM-CKO-200286) with *Csf1r*-CreER (FVB-Tg (Csf1r-cre/Esr1*) 1Jwp/J, Jackson laboratory, stock# 019098) mice. *Tm4sf19*^flox/flox mice without *Csf1r*-CreER were used for wild-type (WT) control. For macrophage-specific TM4SF19 KO induction, tamoxifen (75 mg/kg, Cayman, 13258) dissolved in sunflower oil was treated to *Tm4sf19*^flox/flox/*Csf1r*-CreER mice and WT controls at 6 weeks of age by oral gavage on each of 5 consecutive days. Ten days after the last dose of tamoxifen treatment, mice were used for each experiment. To prolong the duration of Cre recombinase activity in mice, tamoxifen treatment was performed at monthly intervals. The genotyping primers to distinguish between the floxed gene or WT were as follows: forward: 5′- AGGGC AAAGAAGGAAGTGGCTAAT −3′, reverse: 5′-CAGGAAGGGGGCAGACA AGGAGT-3′. The genotyping primers to detect CreER were as follows: forward: 5′-CTTCCAAAGCATGGTCCAGT-3′, reverse: 5′-TGAACCAGC TCCCTATCTGC-3′.

For the diet-induced obesity model, mice at 6 weeks of age were fed a 60% fat diet (HFD; Research Diets, D12492) for up to 12 weeks. A standardized rodent pellet diet (NCD; Purina, 38057) was used for a control chow diet. Energy expenditure, locomotor activities, and food intake were estimated by an indirect calorimetry system (Pheno-Master; TSE Systems). Body composition was measured by nuclear magnetic resonance scanning EchoMRI-700 (Echo Medical Systems).

For the glucose tolerance test, mice were fasted for 12 h before measuring blood glucose levels. Mice were injected with D-glucose (2 g/kg body weights, Sigma, G7021) by intraperitoneal injection. For the insulin tolerance test, mice were treated with insulin by intraperitoneal injection (0.75 units/kg, Sigma, 91077 C). Blood glucose levels were measured from tail vein blood samples at an indicated time using GlucoDoctor Top meter (Allmedicus, AGM-4100) and appropriate glucose indicator strips. For the insulin tolerance test, mice were fasted for 6 h and insulin was injected by intraperitoneal injection. To examine insulin signaling, insulin (0.75 units/kg) was injected into the inferior vena cave of mice that had been anesthetized mice using 2,2,2-tribromoethyl alcohol (avertin; Sigma, T48402) and adipose tissues were harvested 10 min after the injection. To keep the body temperature at 37 °C, we used the heating pad system (Toyotech, DP30-05A). To measure serum insulin, total cholesterol, and triglyceride levels, a mouse insulin ELISA kit (Fujifilm, AKRIN-011T) and chemical blood analyzer (Fujifilm, DRI-CHEM 3500 s) were used.

For the measurement of intestinal lipid excretion, the modified Folch extraction method was used[60]. In brief, lipids from feces (1 g per mouse) were extracted with a chloroform: methanol solution (2:1) and the mass was measured after the evaporation of solvents over 3 days of air drying.

### Bone marrow transplantation

Bone marrow transplantation (BMT) was performed with whole bone marrow cells from WT C57BL/6 N and TM4SF19 KO mice. Bone marrow cells were prepared by flushing the tibia and femur with DMEM[61]. Tail vein injection (2 × 10^6 cells) was performed on lethally irradiated 6-week-old recipient WT mice under anesthesia. After a 4-week recovery period, mice were initiated to be fed an HFD for 12 weeks.

### Human samples

This study adhered to the principles of the Declaration of Helsinki and received approval from the Korea University Guro Hospital (KUGH) Ethics Committee (Institutional Review Board No. 2022GR0095). Prior to participation, all subjects provided written informed consent for the

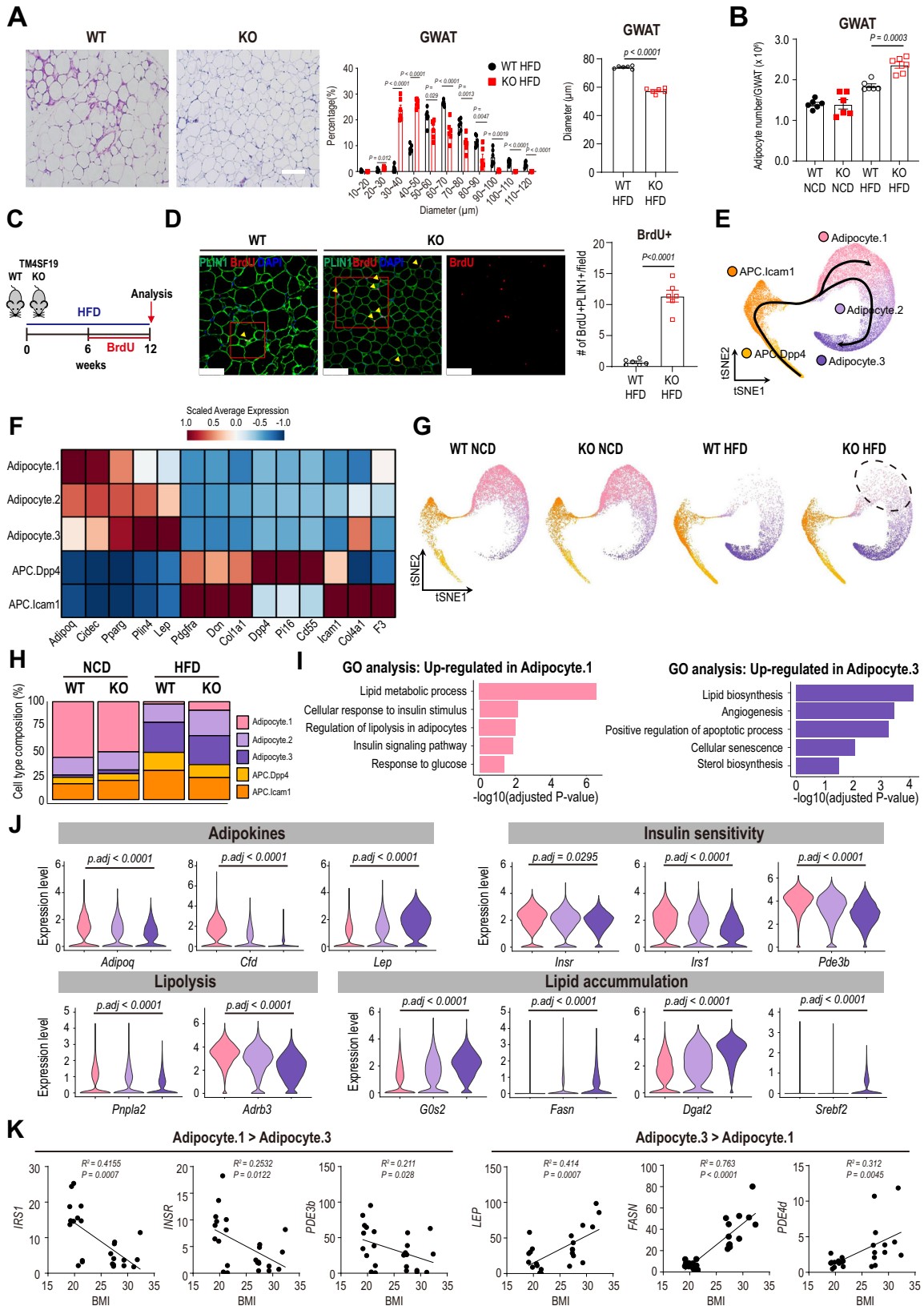

anonymized use of their data for research purposes. The cohort comprised fifteen male and nine female patients, with ages ranging from 9 to 79 years. Human subcutaneous adipose tissue samples were collected, promptly frozen, and stored at −80 °C. Participants of both sexes volunteered for this study, and we did not intend to selectively recruit a particular gender. The sex/gender of participants was

determined through self-report. The characteristics of human fat tissues from 24 individuals are available in Supplementary Table S2.

### Histology
BAT, IWAT, and GWAT were harvested and fixed in 10% formalin (Sigma, HT50128) for 24 h at 4 °C and then embedded in paraffin. 5 μm

**Fig. 6 | TM4SF19 KO restores insulin-sensitive adipocytes in adipose tissue of HFD-fed mice. A, B** Representative images with adipocyte size analysis (**A**), and adipose tissue cellularity analysis (**B**) of GWAT from WT and TM4SF19 KO mice fed HFD (*n* = 6 mice). Scale bars = 50 μm. **C** Schematic diagram depicting BrdU cumulative labeling. **D** Immunohistochemical analysis of BrdU (red) incorporation in PLIN1+ (green) adipocytes in GWAT of WT and TM4SF19 KO mice fed HFD (*n* = 6 mice). Scale bars = 100 μm. **E** tSNE plot showing the differentiation paths from APC to Adipocytes. The color indicates APC and Adipocyte clusters. The line on the tSNE plot shows the differentiation trajectory. **F** Heatmap of canonical marker genes used to identify each cluster. The color indicates the scaled expression level of genes. **G** tSNE plots showing differentiation paths split by condition. The dotted

line (Adipocyte.1) highlights the cluster restored by TM4SF19 KO with HFD. **H** Bar plot showing the composition of APC and adipocyte clusters in each condition. **I** Selected pathways of GO enrichment analysis up-regulated in Adipocyte.1 or Adipocyte.3. **J** Violin plots showing gene expression associated with adipokine secretion, insulin sensitivity, lipolysis, and lipid accumulation. **K** Correlation analysis of Adipocyte.1 and Adipocyte.3 genes in human subcutaneous adipose tissue (*n* = 12 patients). Data are presented as mean values ± SEM. *p*-values were determined by the unpaired two-sided Student's *t*-test (**A, D, J**), two-tailed Pearson correlation (**K**), and two-way ANOVA followed by Bonferroni post hoc tests (**B**). Source data are provided as a Source Data file.

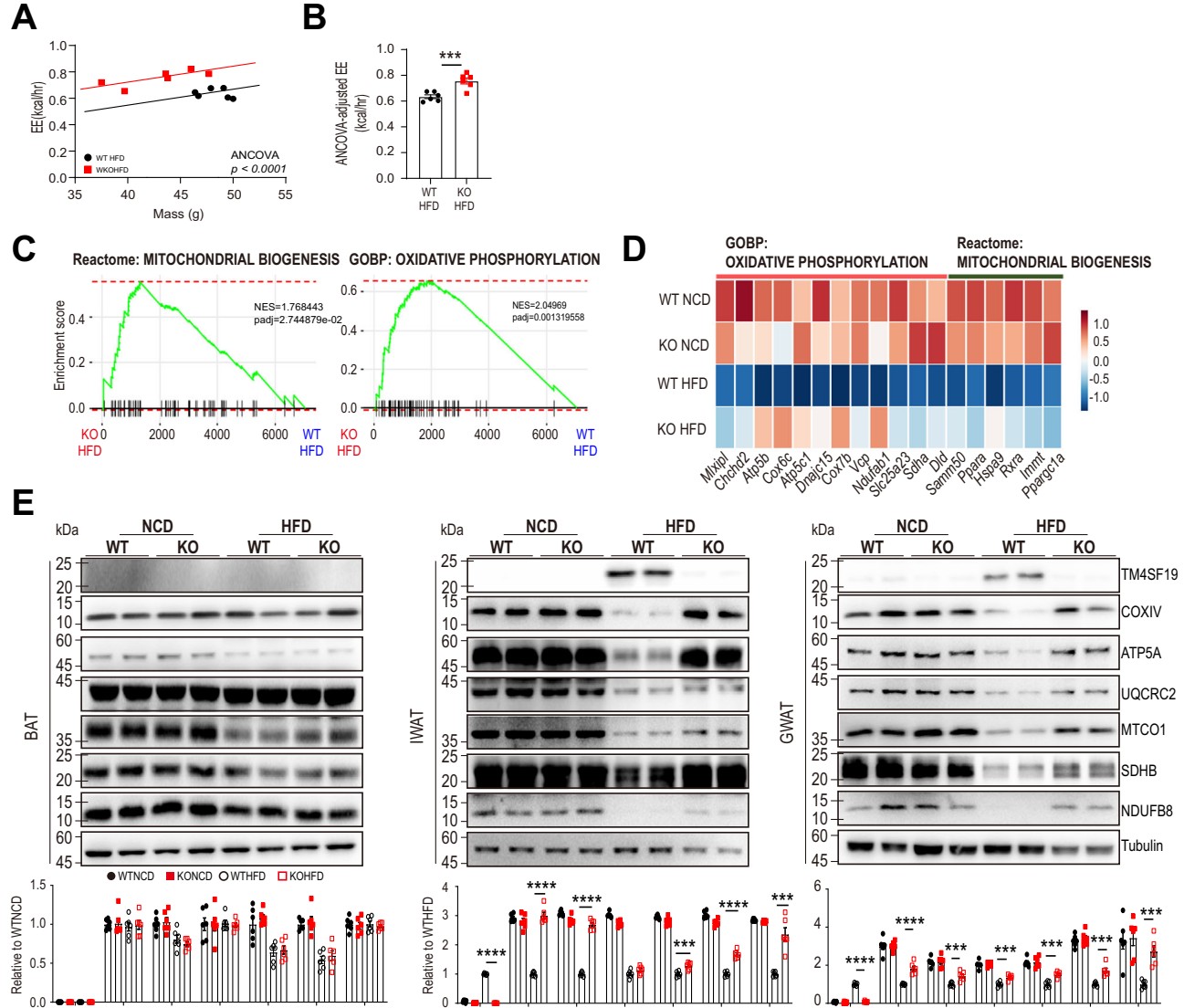

**Fig. 7 | TM4SF19 deletion increased energy expenditure and mitochondrial oxidative metabolism in adipose tissue of mice fed HFD.** WT and TM4SF19 global KO mice were fed HFD for up to 12 weeks. Indirect calorimetry analysis was performed at 10 weeks of HFD feeding. Samples for snRNA-seq and immunoblot analysis were collected at 12 weeks of HFD feeding. **A.** Regression plots of energy expenditure against body mass (ANCOVA using body mass as a covariate, two-sided without adjustment, *n* = 6 mice). **B** ANCOVA -adjusted energy expenditure (EE) (predicted energy expenditure at the mean body mass (45.7 g). *p*-values were determined by the unpaired two-sided Student's *t*-test (*n* = 6 mice). **C** GSEA showing enrichment in mitochondrial biogenesis (NES = 1.768443, adjusted *p*-value = 2.744879e-02) and oxidative phosphorylation (NES = 2.04969, adjusted *p*-value =

0.001319558) by genes significantly contributed to all cells of GWAT of WT HFD and TM4SF19 KO HFD mice. Among the variable genes ordered according to WT HFD and TM4SF19 KO HFD group loading, genes associated with oxidative phosphorylation and mitochondrial biogenesis were marked with a black vertical line. **D** Heatmap analysis of the upregulated genes involved in oxidative phosphorylation, and mitochondrial biogenesis in all cells of GWAT of TM4SF19 KO HFD mice. **E** Western blot analysis of mitochondrial protein levels in TM4SF19 KO mice after feeding HFD for 12 weeks. *p*-values were determined by two-way ANOVA followed by Bonferroni post hoc tests (*n* = 6 mice). Data are presented as mean values ± SEM. Source data are provided as a Source Data file.

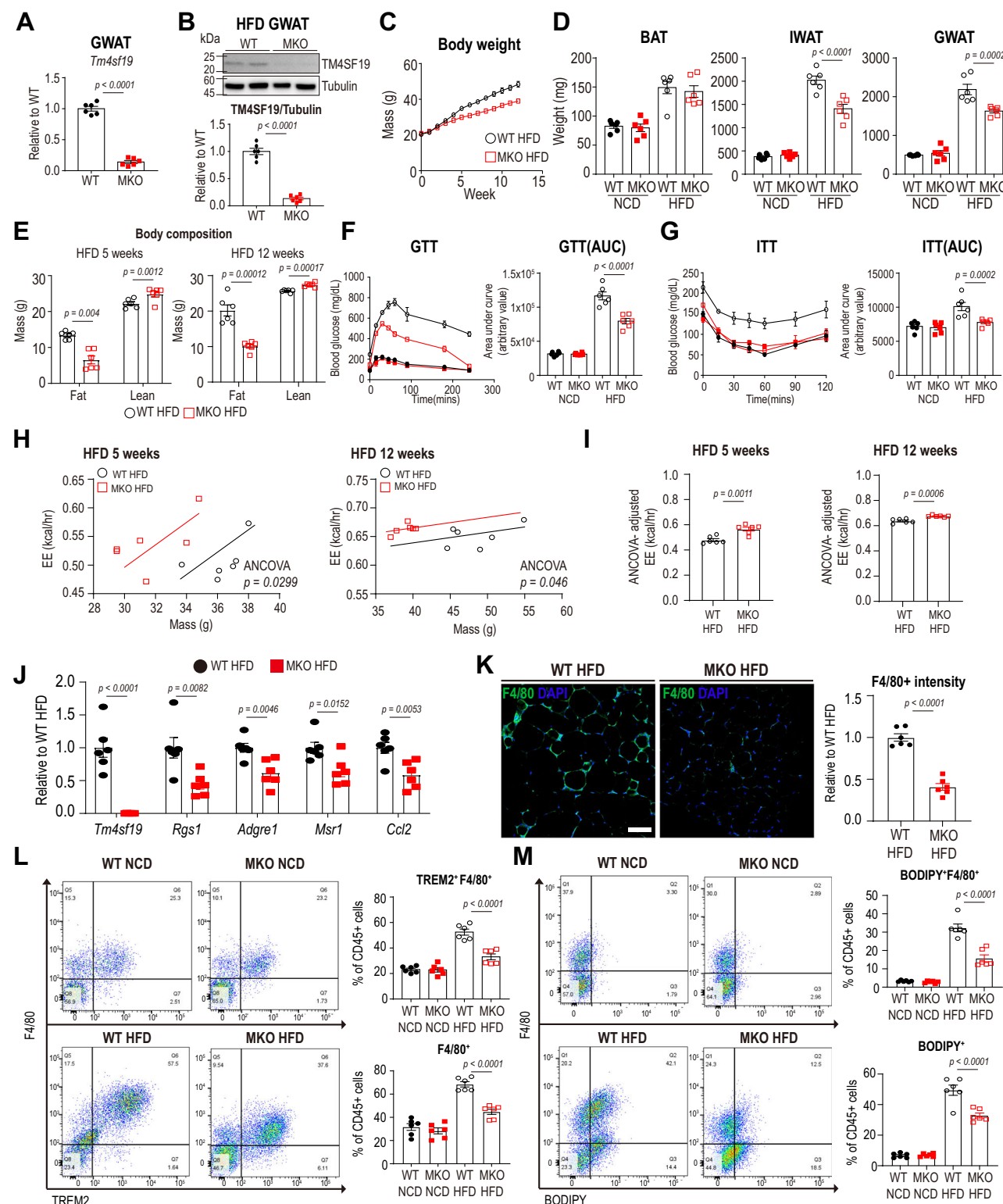

paraffin sections were prepared from the paraffin-embedded tissue blocks. For hematoxylin and eosin (H&E) staining, deparaffinized sections were stained with ClearView Staining Hematoxylin (BBC Biochemical, MA010081) and Eosin Y Alcoholic solution (BBC Biochemical, 3610). The images were obtained with Nikon Elements (NIS BR Analysis ver 5.10.00). For F4/80 staining in liver and GWAT, 5 μm paraffin sections were deparaffinized with $2 \times 10$ min in 100% xylene (Daejung, 1330-20-7), 10 min in 50:50 xylene: ethanol, 5 min in 100% ethanol (Daejung, 4022-4410), 5 min in 95% ethanol, 5 min in 70%

ethanol. After the deparaffinization step, the slides were rinsed with distilled water, boiled for 10 min in citrate buffer (pH 6.0) for antigen retrieval, and cooled to room temperature for 10 min. Blocking was performed with PBS mixed with 3% BSA for 1 h, and F4/80 (30325, 1:200), PDI (3501, 1:200), and LAMP1 (3243, 1:200) was stained with antibody. DAPI (1000x) was stained for 5 min after antibody staining. For each biological replicate of each condition, we randomly selected six areas for each tissue section as technical replicates. The data are presented as average staining intensity values from each condition.

**Fig. 8 | Macrophage-specific TM4SF19 KO improves obesity-induced metabolic dysfunction. A, B** qPCR (**A**) and immunoblot (**B**) analysis to confirm macrophage-specific TM4SF19 deletion from GWAT of tamoxifen-induced *Tm4sf19*[flox/flox] *Csf1r*-CreER (TM4SF19 MKO) mice (*n* = 6 mice). **C, D** Mouse weights monitoring (**C**), adipose tissue weight (**D**) of TM4SF19 MKO mice after HFD feeding for 12 weeks (*n* = 6 mice). **E** Body fat and lean mass of TM4SF19 MKO mice after 5 weeks or 12 weeks of HFD feeding (*n* = 6). **F, G** Glucose tolerance test (GTT) (**F**) and insulin tolerance test (ITT) (**G**) of TM4SF19 MKO mice after HFD feeding for 12 weeks (*n* = 6 mice). **H** Regression plots of energy expenditure against body mass (ANCOVA using body mass as a covariate, two-sided without adjustment, *n* = 6 mice). **I** ANCOVA-adjusted energy expenditure (EE) (predicted at the mean body mass (34.01 g at HFD 5 weeks/43.7 g at HFD 12 weeks)) (*n* = 6). **J** mRNA expression levels of genes involved in inflammatory response from GWAT of TM4SF19 MKO mice after HFD feeding for 12 weeks (*n* = 6 mice). **K** Immunofluorescence staining of F4/80 (green) with DAPI (blue) counterstaining in paraffin sections from GWAT of WT and TM4SF19 MKO HFD-fed mice. Scale bar = 100 μm (*n* = 6 mice). **L, M** Representative flow profiles of TREM2+macrophages and F4/80 expression levels (**L**) and BODIPY+F4/80+ macrophages and BODIPY+CD45+ cells (**M**) from GWAT of TM4SF19 MKO HFD-fed mice (*n* = 6 mice). Data are presented as mean values ± SEM. *p*-values were determined by the unpaired two-sided Student's *t*-test (**A, B, E, I–K**) and two-way ANOVA followed by Bonferroni post hoc tests (**D, F, G, L, M**). Source data are provided as a Source Data file.

Images were acquired with a Zeiss confocal microscope (LSM800) and analyzed with Zen software (version 3.0).

## BrdU labeling and adipocyte cellularity analysis

To achieve cumulative labeling with BrdU, mice were fed a high-fat diet for 6 weeks, followed by administration of 0.8 mg/ml BrdU (B5002, Sigma) in drinking water, refreshed every 2 days while continuing the HFD regimen. GWAT was collected after 6 weeks of BrdU administration (12 weeks after HFD feeding). For immunohistochemical analysis, anti-bromodeoxyuridine-fluorescein (Roche, 11202693001, 1:50), PLIN1 (Everest, EB07728, 1:300) antibodies, along with Goat anti-Rabbit Alexa Fluor™ 594 (Invitrogen, A-11012, 1:400) antibodies, were employed. Total adipocyte numbers were estimated using a method detailed in previous studies[42].

## Western blot and quantitative PCR

To perform western blot analysis, BAT, IWAT, and GWAT depots from WT and TM4SF19 KO mice were homogenized in PRO-PREP Protein Extraction Solution (iNtRON Biotechnology, 17081) containing SIG-MAFAST Protease Inhibitor Cocktail (Sigma, S8820) and PhoSTOP phosphatase inhibitors (Roche, 4906845001) using tacoPrep Bead Beater homogenization system. The protein concentration of protein samples was quantified by BCA assay and measured by spectrophotometry (MultiSkan GO, 51119000, Thermo Fisher Scientific) at 562 nm using SkanIt software ver 5.0. 10 μg protein samples were denatured in 5 x sample buffer (Elpis Biotech, EBA 1052) in 95 °C for 5 mins and separated on 8% or 12% of SDS-PAGE gel and transferred to PVDF membrane (Bio-Rad). The membrane was blocked with 5% non-fat dry milk or BSA in TBST (Tris-buffered saline with 0.1% Tween 20) and incubated with primary antibody at 4 °C overnight followed by horseradish peroxidase (HRP)-conjugated secondary antibody incubation at room temperature for 1 h. Anti-rabbit TM4SF19 antibody (YT6290, WB 1:1000) was purchased from Immunoway. Anti-rabbit α/β-Tubulin (2148, WB 1:2000), anti-rabbit NF-κB p105/50 (13586, WB 1:1000), anti-rabbit ATP6V1B2 (14617, WB 1:1000), anti-rabbit LAMP1 (3243, WB 1:1000), anti-rabbit F4/80 (30325, IHC 1:200), anti-rabbit p-IRS1 (S612) (3203, WB 1:1000), anti-rabbit IRS1 (3407, WB 1:1000), anti-rabbit COXIV (4850, WB 1:1000), anti-rabbit Myc-Tag (2278, IP 1:200), and normal rabbit IgG (2729, IP 1:100) were purchased in Cell Signaling. Anti-rabbit ATP6V0B (NBP2-83943 WB 1:1000) was purchased in Novus Biological. Anti-rabbit ATP6V1A (GTX110815, WB 1:1000) was purchased in GeneTEX. Anti-mouse p-AKT (S473) (Sc-514032, WB 1:1000) and anti-mouse AKT (Sc-81434, WB 1:1000), anti-mouse ATP6V0D2 (sc-517031, WB 1:500), anti-mouse SREBP1 (Sc-13551, WB 1:1000) were purchased from Santa Cruz. Anti-mouse total oxphos cocktail (Ab110413, WB 1:1000) was purchased from Abcam.

For quantitative PCR (qPCR) analysis, total RNA from fat tissues from mice and humans was isolated by TRIzol reagent (Thermo Fisher Scientific, 15596018) according to the manufacturer's protocols. High-Capacity cDNA Reverse Transcription kit (Applied Biosystems, 4368814) was used to synthesize cDNA, and qPCR reaction was performed with iQ SYBR Green Supermix (Bio-Rad, 170-8884) in Bio-Rad CFX Connect Real Time PCR Detection system. *β-actin* and peptidylprolyl isomerase A (*PPIA*) were used as a housekeeping gene and primers used for qPCR are listed in Supplementary Table S3. 2-ΔCt method was used to calculate the relative expression levels of each gene.

## Flow cytometric analysis

GWAT from HFD-fed WT and TM4SF19 KO mice, were used for flow cytometric analysis. Adipose tissues were isolated and digested with filtered 2 mg/mL collagenase type I (Gibco, 17100-017) in KRBB buffer containing 3% BSA at 37 °C. Fully dissociated preparations were then passed through a 100 μm cell strainer and centrifuged at 500 × *g* for 5 min. Pellets were then washed with DMEM containing 20% FBS two times. Stroma vascular fraction (SVF) stained with fluorescence-labeled appropriate primary antibodies at room temperature. BV421 Rat Anti-Mouse F4/80 (565411, 1:50) was purchased from BD Biosciences. Human/Mouse TREM2 Allophycocyanin Mab (FAB17291A, 1:50) were purchased from R&D systems. APC/Cyanine7 anti-mouse CD45 (103116, 1:100), FITC anti-mouse/human CD11B (101206, 1:100), BV711 anti-mouse/human CD11B (101242, 1:100), Brilliant Violet 421™ anti-human CD206 (141717, 1:100), PE/Cyanine7 anti-mouse CD11C (117317, 1:100), PE anti-mouse/human LYVE1 (12-0443-82, 1:50) were purchased from Biolegend. FITC anti-mouse CD45 (11-0451-82, 1:100) was purchased from Invitrogen.

Unstained and single-stained controls were used for compensation. LSRFortessa X-20 Flow Cytometer (BD Biosciences) was used to analyze the samples and the data was acquired by BD FACSDiva 8.0 software which then was further analyzed by Flowjo software (version 10.5.3; TreeStar). The gating strategies are provided in Supplementary Fig. 12.

## Adipose tissue fractionation and magnetically activated cell sorting (MACS)

Fully dissociated GWAT as described above were centrifuged at 300 × *g* for 5 min. Red blood cells were lysed with RBC lysis buffer (15.5 mM NH4Cl, 1.2 mM NaHCO3 and 50 mM EDTA, pH 7.4) and pellets (SVF) were then washed with PBS with 3% BSA (500 × *g* for 5 min). SVF were incubated with anti-F4/80-FITC antibody (Biolegend) for 2 h at 4 °C followed by anti-FITC-microbeads (Miltenyi Biotec, 130-048-701, 1:10) incubation for 1 h at 4 °C. After incubation, F4/80+ cells were isolated by column (Miltenyi Biotec, 130-122-727) according to the manufacturer's instruction (Miltenyi Biotec, MACS technique).

## Two-photon intravital imaging of adipose tissue in mice

The imaging system utilized for this study consisted of an Olympus FVMPE-RS Two-photon microscope with fluorescent signals being detected through FV30-FGR and FV30-FVG filters. When conducting adipose tissue imaging, mice were placed under anesthesia using zoletil, and GWAT was meticulously exposed. The mice were then positioned within a chamber and kept warm with the aid of a heating pad to maintain body temperature during intravital imaging. Intravital imaging was performed for the period of about 6-7 h and sequential images were captured at intervals of either 1 min 30 s or 2 min. To quantify the crown-like structures, an extensive series of images were

captured and analyzed using Volocity (Improvision, Germany) and Imaris software (Bitplane, Switzerland). Furthermore, to monitor macrophage migration, GFP-positive cells within tissue sections were tracked, and mobility metrics including velocity, displacement, and meandering index were computed using Volocity, Imaris software, and MATLAB software. Velocity, displacement, and meandering index were analyzed based on 6 h-length of two-photon intravital imaging of adipose tissue in each mouse. The experiments were independently performed more than 3 times in each condition. Macrophages were collected from all intravital imaging and analyzed together.

To visualize the vessel structure in GWAT, CF®405M-conjugated wheat germ agglutinin (WGA; Biotium, 29028, 2.5 mg/kg) was delivered by tail vein injection. Using anesthesia cream, the GWAT was opened surgically and was appropriately fixed between the bottom surface and cover slip attached to the custom-made chamber to prevent motion artifacts during imaging. The chamber was maintained at 37.0 °C to keep the body temperature by the heating pad system (Toyotech, DP30-05A). Mice were continuously anesthetized using a mixture of 1% isoflurane (Hanapharm, 1045) and 0.7 liter/min oxygen. Real-time imaging was performed by the FV1000MPE Two-photon system (Olympus) at 800 nm for green (CX3CR1) and blue (WGA). The imaging depth was 60μm from the surface of GWAT with a complete Z-stack. One-minute interval time-lapse image acquisition was quantified until about 6 h. WGA was injected (i.v.) to visualize blood vessels during two-photon intravital imaging. The WGA blue in blood vessels was frequently engulfed by circulating phagocytes such as monocytes and neutrophils, which resulted in partial blue-labeling of these cells. Therefore, the extravasated phagocytes to adipose tissues were partially visualized with blue, which was represented in red in these snapshots and the supplementary video.

## Cell culture

Dulbecco's Modified Eagle Medium (DMEM; Welgene, LM001-07) supplemented with 10% fetal bovine serum (FBS; Gibco, 16000044) and 1% penicillin/streptomycin (PS; Welgene, LS202-02) were used for growth medium. RAW264.7 cells (ATCC, TIB-71) were maintained in a growth medium until experiments were initiated. To check the localization of TM4SF19, we used LysoTracker (Invitrogen, L7525) and MitoTracker (Invitrogen, M7512) following manufacturer's protocols. For colocalization analysis, the Pearson correlation coefficient was measured by JACOP in ImageJ software (version 1.52a).

C3H10T1/2 cells (ATCC, CCL-226) were used to conduct in vitro adipocyte experiments. For adipogenic lineage commitment, C3H10T1/2 cells were exposed to 20 ng/mL bone morphogenetic protein 4 (BMP4) (R&D systems, 314-BP) in the growth medium for 3 days. For adipogenic differentiation, the cells were incubated with the differentiation medium [growth medium containing 0.125 mmol/L indomethacin (Cayman Chemical, 70270), 2.5 mmol/L isobutyl methylxanthine (Cayman Chemical, I5879), 1 μmol/L dexamethasone (Cayman Chemical, D4902), 10 μg/mL insulin, and 1 nmol/L triiodothyronine (Cayman Chemical, 16028)] for 3 days. Then the cells were maintained with the growth medium containing 10 μg/mL insulin and 1 nmol/L triiodothyronine until the experiments were performed.

Bone marrow-derived macrophages (BMDMs) were extracted from femur and tibia of 6 weeks WT and TM4SF19 KO mice[61]. Briefly, the bone marrows from the femur and tibia were washed and resuspended by injecting 2 mL of DMEM and centrifuge bone marrow suspension at $500 \times g$ for 5 min at 4 °C. Bone marrow was resuspended in 1 mL of RBC lysis buffer and centrifuge at $500 \times g$ for 5 min at 4 °C. For BMDM differentiation, bone marrow cells were incubated with bone marrow culture medium [growth medium with 1 ng/mL M-CSF1 (Peprotech Ltd, 315-02)] for 3 days at 37 °C, 5% $CO_2$. On the third day, the media was changed with fresh bone marrow culture medium for an additional 3 days of differentiation. LysoSensor Yellow/Blue dextran (Invitrogen, L22460) staining was performed to measure

lysosomal pH following the manufacturer's protocols. 1 mg/mL of LysoSensor Yellow/Blue dextran was incubated for 24 h, and 4% PFA was incubated for 10 min for fixation. Images were acquired with Zeiss (LSM800) and analyzed with Zen software (version 3.0).

## Subcellular fractionation

To collect the membrane and cytosolic fractions, we performed subcellular fractionation using the method described previously[62]. A total of $2 \times 10^8$ cells were lysed and homogenized using a fractionation buffer. The homogenates were then centrifuged at $500 \times g$ for 10 min to clear the nuclei and intact cells. The resulting supernatant was collected and subjected to further centrifugation at $100,000 \times g$ for 4 h to isolate the cytosolic (supernatant) and membrane (pellet) fractions. The resulting pellet was lysed with RIPA buffer (Thermo Fisher Scientific, 89901). To confirm the purity of the isolated membrane and cytosol fractions, two marker proteins, LAMP1 and β-actin, were used for membrane and cytosolic fraction markers, respectively.

## Lysosome isolation

Lysosome isolation was performed using a previously described method[63]. A total of $2 \times 10^8$ cells were rinsed and collected with a fractionation buffer. The samples were then centrifuged at $1000 \times g$ for 10 min and collected the resulting supernatant. The supernatant was centrifuged at $20,000 \times g$ for 20 min and the resulting pellets were mixed with 19% diluted Optiprep density gradient medium (Biovision, M1248). The mixture was then loaded onto a low osmotic discontinuous density gradient with 27%, 22.5%, 19%, 16%, 12%, and 8%[63] and centrifuged at $150,000 \times g$ for 4 h. Subcellular fractions were eluted with RIPA lysis buffers (Thermo Fisher Scientific, 89901). To verify the purity of the lysosome fractions, the lysosome marker LAMP1 was tested in individual fractions by immunoblot analysis. It was found that fraction 2 (22.5%) was enriched with the lysosome. High-purity lysosome fractions (fraction 2) were used to measure V-ATPase activity by ATPase activity assay (Abcam, ab234055) following the manufacturer's protocols. Briefly, 50 μg of fresh lysosomal proteins were incubated with 5 mM sodium azide (mitochondrial ATPase inhibitor) in the presence or absence of concanamycin A (ConA: specific inhibitor of V-ATPase) for 30 min at 37 °C. The V-ATPase activity was normalized with experimental values in the absence of ConA and was then subtracted with control values in the presence of ConA.

## Gene overexpression

Lentiviral transfer plasmid *Tm4sf19* ORF was generated by inserting *Tm4sf19* (NM_001160402.1) ORF clone to the transfer vector pLenti-EF1a-C-mGFP-P2A-Puro lentiviral vector (Origene, PS100121). Myc-tagged transfer plasmid was generated through GFP-fused TM4SF19 by replacing mGFP with myc-DDK. Lentivirus was produced by transfection of HEK293T cells (ATCC, CRL-3216) with transfer plasmids (5 μg), psPAX2 (3.75 μg, Addgene, 12260), and pMD2.G (1.25 μg, Addgene, 12259) in 4:3:1 ratio using jetPRIME transfection reagent (Polyplus, 114-15). The medium containing viral particles was collected after 48 h of transfection, centrifuged, and filtered through a 0.45-nm syringe filter (Sartorius, STR.16555 K). RAW264.7 macrophages were transduced with the viral supernatants containing 8 μg/mL polybrene (Santa Cruz, sc-134220) for 72 h and were replaced with growth medium containing 2 μg/mL puromycin (Sigma, P8833) for selection for 48 h.

## Immunoprecipitation

For immunoprecipitation analysis, RAW264.7 macrophages overexpressing Myc-fused TM4SF19 were harvested by lysis buffer (50 mM Tris-HCl (pH 7.6), 150 mM NaCl, 2 mM EDTA, and protease inhibitor). After 10 min of incubation on ice, lysates were centrifuged at 13,000 rpm, at 4 °C for 15 min and the pellet was discarded. 50 μL of

Pierce™ Protein A/G Magnetic Beads (Invitrogen, 88802) were incubated with 5 µL of Myc and IgG antibodies for 2 h rotating at 4 °C. Beads were washed once in lysis buffer before incubation with 300 µg of lysate for 4 h, rotating at 4 °C. Beads were then washed five times in a buffer containing Tris-buffered saline (TBS) containing 0.05% Tween-20 Detergent. Bound proteins were eluted by 0.1 M glycine (pH 2.0).

## Phagocytosis assay
For phagocytosis analysis, fully differentiated C3H10T1/2 adipocytes were treated 5 µg/ml brefeldin A (BFA; Biolegend, 420601) for 24 h and were stained with 4,4-difluoro-5-(2-thienyl)−4-bora-3a, 4a-diaza-s-indacene3-dodecanoic acid (BODIPY 558/568 C12) (Invitrogen, D3835), and macrophages were labeled with Vybrant DiO Cell-Labeling Solution (Invitrogen, V22886) overnight. BODIPY-labeled C3H10T1/2 cells were detached, added to the DiO-labeled macrophages in a 1:1 ratio. Live cell imaging was performed with the Operetta CLS High-Content Analysis System (Perkin Elmer) and analyzed by Harmony High-Content Imaging and Analysis Software (Perkin Elmer). For detection of high-resolution images of phagocytosis of dying adipocytes by the macrophages, fully differentiated adipocytes were treated with 5 µg/ml BFA(Biolegend) for 24 h and then co-cultured with BMDM stained by Alexa488-cholera toxin B (CtB, Invitrogen, C22841) for 90 min. After co-culturing, fixation was performed with 4% PFA for 15 min and then LipidTOX (Invitrogen, H34476) was stained for 1 h according to the manufacturer's protocols. Images were acquired with Zeiss (LSM800) or Leica (TCSSP8) confocal microscope and analyzed with Zen software (version 3.0) or Leica software (LASX).

## Promoter analysis
CiiiDER, an integrated computational toolkit for transcription factor binding analysis, was performed to analyze the promoter region (2.5 kb) of mouse *Tm4sf19* and human *TM4SF19* gene (ENSMUSG00000079625, ENSG00000145107). Only match scores of 0.85 or above were accepted to retrieve the accurate transcription factor binding sites. For the chromatin immunoprecipitation (ChIP) analysis, the MAGnify Chromatin Immunoprecipitation System kit (Invitrogen, 49-2024) was used. RAW264.7 macrophages were treated with LPS for 4 h or oxLDL (100 µg/ml, Invitrogen, L34357) for 24 h before crosslink. The macrophages were crosslinked with 1% formaldehyde for 10 min at room temperature and the reaction was stopped by adding 1.25 M glycine. Immunoprecipitation was performed using the antibody against NF-κB1 p105/p50 (Cell Signaling, 13586) and SREBP1 (Santa Cruz, sc-13551) as a negative control. To perform ChIP-qPCR analysis and gel-electrophoresis, precipitated DNA fragments were analyzed by specific primers for the putative NF-κB and SREBF1 binding site bearing promoter region of TM4SF19 (NF-κB Forward: 5′- AGGGGTGCTCTCTCAGAAG-3′ and NF-κB Reverse: 5′-GCTCAACGGTCTTAGT-3′; SREBF1 Forward: 5′- CTTCCATTGAAAT-GATCATATGATTCTTG-3′ and SREBF1 Reverse: 5′-CCAGGGCTACACA-GAGAAAC-3′).

## Single-nucleus RNA sequencing
Tissues were incubated in TST lysis buffer (146 mM NaCl, 10 mM Tris-Cl pH7.5, 1 mM CaCl$_2$, 21 mM MgCl$_2$, 0.03% Tween-20, 0.01% BSA) for 10 min at 4 °C and minced using Noyes spring scissors. Samples were filtered through a 70 µm strainer and centrifuged at 500 × *g* for 5 min at 4 °C. Pellets were washed once with TST lysis buffer and twice with 1% BSA/PBS. Pelleted nuclei were resuspended in 1% BSA/PBS with 0.2U/µl RNase inhibitor and filtered through a 30 µm strainer. Nuclei were stained with acridine orange/propidium iodide solution and counted using an automated counter (LUNA-FX7, Logos Biosystems, L70001).

For sample multiplexing, 1-2 × 10$^6$ nuclei per sample were stained with 100 µl of CellPlex (10X Genomics, PN-1000261). Samples were washed three times with 2 ml of cold 1% BSA/ PBS and centrifuged at 500 × *g* for 5 min at 4 °C. Nuclei were resuspended in 1% BSA/PBS and

equal numbers of cells from each biological replicate for a condition were mixed. Nuclei were counted and added to the reverse transcription mixture to capture 20,000 nuclei of the combined sample.

Libraries for single-nucleus RNA sequencing were generated using the Chromium Next GEM Single Cell 3'Kit (10X Genomics, PN-1000269), 3'Feature Barcode Kit (10X Genomics, PN-1000262), Dual Index Kit TT Set A (10X Genomics, PN-1000215), and Dual Index Kit NN Set A (10X Genomics, PN-1000243). To generate gel bead-in-emulsions (GEMs), samples were loaded onto the Chromium Next GEM Chip G (10X Genomics, PN-2000127), and the subsequent microfluidic process was performed on a Chromium controller (10X Genomics). Reverse transcription was performed by incubating GEMs in a thermal cycler (C1000 Touch Thermal Cycler with Deep Well, Bio-Rad). After incubation, GEMs were disrupted and pooled cDNA was purified using silane magnetic beads. Amplification of cDNA was performed using a thermal cycler. During cDNA size selection using SPRIselect beads, the supernatant and bead pellet portion were handled separately to obtain cDNA from poly-adenylated mRNA and DNA from cell multiplexing oligo feature barcode. To generate cell multiplexing libraries, 75 µl of supernatant was retained and further purified. The cell multiplexing barcode DNA and indexing primers (10X Genomics, PN-3000482) were added to the PCR reaction mixture and PCR amplification was performed. For cDNA library generation, bead pellets were purified and cDNA was eluted. Subsequent library construction procedures, including enzymatic fragmentation, end repair, A-tailing, adaptor ligation, and sample index PCR, were performed according to the manufacturer's protocol.

The size and concentration of cDNA and libraries were analyzed using a Bioanalyzer (Agilent). Libraries were pooled and sequenced on a HiSeq X Ten sequencer (Illumina) using a 100 bp paired-end protocol to generate a minimum of 20,000 read pairs per cell for the 3'gene expression library and 5000 read pairs per cell for the cell multiplexing library.

## Single-nucleus RNA sequencing data pre-processing
Raw reads from multiplexed single-nucleus RNA sequencing data were aligned to the mouse reference genome GRCm38 and demultiplexed by sample-specific barcodes using cellranger (v6.1.1). The raw count matrix for each sample was generated using cellranger multi with min-assignment-confidence = 0.7, expect-cells = 20,000, and include-introns = true options. R (v4.1.0) was used for analysis. Low-quality cells with detected UMIs lower than 1000 or a percentage of genes mapped to mitochondrial genes higher than 5% were filtered out for further analysis. The threshold was determined by visually checking the distribution of per-cell UMI count using addPerCellQC function of scater (v1.22.0) R package[64]. All count matrices were combined and cells were clustered using quickCluster function of scran (v1.22.1) R package[65]. Cell-specific size factors were computed using compute-SumFactors function of the same package. The normalized count matrix was generated with logNormCounts function using the calculated size factors with pseudo_count = 1. To select highly variable genes (HVGs), variance of normalized expression for each gene was modeled using modelGeneVar and getTopHVGs function of scran R package with fdr.threshold = 0.05. The dataset was scaled and centered using ScaleData function of Seurat (v4.0.5)[66]. The top 25 principal components (PCs) are computed on HVGs and used for clustering and UMAP dimension reduction. To identify the molecular characteristics of each cluster, marker genes of each cluster were obtained using FindAllMarkers function of Seurat. Cell types were manually annotated in the resolution of clusters based on identified molecular characteristics. Putative doublets were confirmed using scDblFinder (v1.8.0) R package[67]. These cells were manually double-checked whether they expressed two or more cell type markers from different lineages and removed. Cells expressing erythroid cell markers such as *Hba-a1, Hba-a2* and Alas were removed. There were some clusters presumed to be

derived from epididymis, anatomically close to the GWAT. These cells were annotated as epididymal cells (*Abcb5, Ces5a, Adam7*), spermatozoas (*Dnah12, Spef2, Hydin*) and efferent duct cells (*Adcy8, Dnah5, Slc9a3*) and removed for further analysis. The remained cells after quality control were re-clustered using the same functions described above with the top 15 PCs. The batch effect caused by different samples was corrected by RunHarmony function of harmony (v0.1.0) R package[68].

**Single-nucleus RNA sequencing data analysis**

**Trajectory analysis.** Pseudotime analysis was conducted using Palantir (v1.0.0) python package for monocyte/macrophage and APC/Adipocyte cell types[69]. To construct a pseudotime trajectory on tSNE plot, the variable features were selected using getTopHVGs function and a renormalized count matrix of intended populations was extracted. For monocyte and macrophage lineage trajectory, the start_cell was chosen randomly in the monocyte cluster. The first 150 PCs were used to make a diffusion map using run_diffusion_maps with n_components = 10 and determine_multiscale_space function with the default options. Pseudotime and differentiation trajectory were obtained using the run_palantir function with num_waypoints = 2000. The coordinates of tSNE on diffusion components were computed using run_tsne function with perplexity = 700. The constructed trajectory was visualized on tSNE plot using plot_palantir_results function. After that, to find the more reliable starting cell, the cell that has the lowest pseudotime was selected as a start_cell and the above process was repeated. For adipose lineage trajectory, the start_cell was designated as one of the cells expressing *Dpp4* and *Cd34* (Fig. S8A). Trajectory was constructed following the same methods as mentioned above except for these options: the number of used PC = 100, num_waypoints = 5000 for run_palantir function and perplexity = 900 for run_tsne function. The start_cell was selected again as described above.

Monocle3 (v1.3.1) was used for confirmation of monocyte to macrophage trajectory[70,71]. We used the same HVGs and normalized count matrix of monocyte and macrophage populations as used in Palantir. We preprocessed the data using preprocess_cds function with 20 PCs and reduced dimensions using UMAP. Clustering was performed using cluster_cells function with a resolution of 0.0001. The trajectory was constructed using learn_graph with default parameters.

**Cell type composition analysis.** To calculate the cell type composition, the number of cells in each cell type for each sample or condition was divided by the total cell number of each sample. The total count of the cells in each sample was scaled to 100%. The significance was determined by MASC (v0.0.0.9000) R package[72] and the batch was considered a random effect.

**Differential expression and GO analysis.** For publicly available transcriptomic database analysis, differentially expressed genes (DEGs) between different body fat conditions (NCD vs HFD and lean vs obese) were demonstrated by iDEP 9.6[73] and GEO2R provided by NCBI. The significant DEGs of top 100 upregulated genes in each dataset were determined by adjusted *p*-value < 0.05 and fold change >2. DAVID was used for functional enrichment analysis[74]. DEGs were determined between Adipocyte.1 and Adipocyte.3 using limma (v3.50.0) R package[75]. Briefly, lmFit function was used to fit the model on the normalized count for each gene and batch effects were considered using duplicateCorrelation function by adding sample information to the block option. Statistical test was conducted using eBayes function with trend=T and robust=T options. The significant DEGs (adjusted *p*-value < 0.05) between Adipocyte.1 and Adipocyte.3 were used as input. The DEGs up-regulated in Adipocyte.1 and up-regulated in Adipocyte.3 were separately used for each cluster-enriched pathway analysis.

For gene set enrichment analysis (GSEA), DEGs between the WT HFD and TM4SF19 KO HFD conditions were obtained using FindMarkers function of Seurat R package and Molecular Signatures Database (MSigDB) was downloaded using msigdbr (v7.4.1) R package. DEGs were sorted by average log2 fold change and applied for GSEA using fgsea (v1.21.2) R package.

**Calculating signature score.** *Trem2*+ macrophage and *Lyve1*+ macrophage signature scores were calculated using AddModuleScore function of Seurat. Up-regulated DEGs were used as signature genes. DEGs were determined between *Lyve1*+macrophage and *Trem2*+macrophage in wild-type mice (WT NCD, WT HFD), filtered with adjusted *p*-value < 0.05. Mac1 and Mac3 signature scores were computed using same function. The DEGs between Mac3 (*Trem2*-expressing macrophage) and Mac1 (*Lyve1*-expressing macrophage) were obtained from the previous study[4] and filtered with *p*-value < 0.05 and log2fc >1.

**Co-expression network analysis.** The co-expression gene modules were obtained using hdWGCNA (v0.2.04) R package, which was applied after sub-clustering the monocyte and macrophage population of the WT HFD condition. 1837 metacells were constructed using the MetacellsByGroups function with $k = 25$ parameter and genes expressed in at least 5% of the cells used for downstream analysis. Co-expression gene network was constructed using the ConstructNetwork function with soft power threshold of 3, chosen using the TestSoftPowers function. DEGs between WT HFD and TM4SF19 KO HFD conditions of monocyte and macrophage population were obtained using FindMarkers function of Seurat R package, and down- or up-regulated genes were determined with adjusted *p*-value < 0.05 and | log2fc | > 0.2.

**Macrophage module score analysis.** Macrophage from the WT HFD condition were subset and processed as described above and visualized on a PCA plot with PCs obtained using RunPCA function of Seurat R package. The cell order along the trajectory from Trem2- to Trem2+ macrophages was determined using the infer_trajectory function of SCORPIUS (v1.0.9) R package. The module scores of each biological term - phagocytosis (*Stab1, Cd163, Mertk, Mrc1*), cholesterol efflux (*Abca1, Apoe*), lipid handling (*Plin2, Pparg, Lipa*), vacuolar acidification (*Atp6v1h, Atp6v0d2, Atp6v1b2, Atp6v0a1, Atp6v1a*) were obtained using AddModuleScore function of Seurat R package and were smoothed along the ordered cells using sliding window size equal to 20% of the total number of cells.

**Statistics**

GraphPad Prism 9 software (GraphPad Software, USA) was used for statistical analysis and correlation analysis. Data are presented as mean ± standard errors of the mean (SEM) or mean ± standard deviation (SD) as indicated in Figure Legends. National Institutes of Health ImageJ software (version 1.52a) was used to quantify the intensity of immunoblot and immunostaining.

**Reporting summary**

Further information on research design is available in the Nature Portfolio Reporting Summary linked to this article.

## Data availability

The raw snRNA-seq data generated in this study have been deposited in the NCBI Sequence Read Archive (SRA) database under accession code PRJNA942977. The publicly available human and mouse adipose tissue data used in this study are available in the gene expression omnibus (GEO) database under accession code GSE59034, GSE150102 and GSE182930. The publicly available human white adipose tissue sc/ snRNA-seq data is available under accession code GSE176171. Mouse reference genome GRCm38 (mm10) was used to align raw reads. All data generated in this study are provided in the Article, Supplementary Information, and Source Data file. Source data are provided with this

paper. Other data supporting the findings of this study are available from the corresponding authors on request. Source data are provided with this paper.

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

## Acknowledgements

This research was supported by the National Research Foundation of Korea (NRF) grants (RS-2023-00213572 and NRF-2018R1A5A2024425 to Y.-H.L., NRF-2021M3H9A1030158 and 2021R1A6A1A10042944 to J.K.K., RS-2023-00207834 to Y.-M.H., 2013M3A9D5072550 to J.K.S.) funded by the Ministry of Science and ICT (MSIT) of the Korean Government and Korea University Guro Hospital grant (KOREA RESEARCH-DRIVEN HOSPITAL) funded by Korea University Medicine (No. K2210441). J.G.G. was supported by US National Institutes of Health grant (R01DK062292).

## Author contributions

Y.-H.L. and J.K.K. conceived and designed the study. C.C., M.K., H.K., S.L., S.K., H.J., J.K.S., and Y.-S.J. conducted the animal experiments. Y.L.J., Y.H.C., and J.K.K. conducted snRNA-seq analysis. C.C. and M.K. performed in vitro experiments. S.N. provided human samples. S.N. and C.C. analyzed human samples. C.C., M.K., G.C., and Y.C. conducted bone marrow transplantation experiments. K.-M.P., C.C., M.K., and Y.-M.H. conducted long-term intravital imaging experiments. Y.-H.L., C.C., Y.L.J., J.K.K. and J.G.G. wrote the manuscript. All authors reviewed the manuscript.

## Competing interests

The authors declare no competing interests.
