## [Peer Review File · Nature Communications]

TM4SF19-mediated control of lysosomal activity in macrophages contributes to obesity-induced inflammation and metabolic dysfunctionREVIEWER COMMENTS

Reviewer #1 (Remarks to the Author):

Choi et al identify enhanced expression of Tm4sf/TM4SF19 in obese adipose tissue of mice and man. They identify TM4SF19 to be expressed by macrophages strongly induced upon HFD. Using elegant in vitro studies, they identify TM4SF19 as an interaction partner of ATP6V0B preventing the assembly of a lysosomal V-ATPase and thus regulating lysosomal acidification. They show that in BMDM loss of TM4SF19 reduced lysosomal acidification and enhanced the clearance of dying adipocytes. In vivo, they find that loss of TM4SF19 shifts macrophage and adipocyte dynamics overall resulting in reduced body weight gain, enhanced glucose and insulin sensitivity and reduced inflammatory responses.

The paper is of particular interest as it addresses the fine-tuning of tissue remodeling processes by macrophages in the context of obesity, which might contribute to identifying potential therapeutic targets for the treatment of obesity and associated disorders. Using elegant in vitro and in vivo models as well as state of the art sequencing technologies the authors identify a novel regulator of macrophage biology and uncover a strong phenotype depending on the lysosomal protein TM4SF19. However, I have some remarks which need to be addressed by the authors:

1. Besides the characterized NF κ B binding site in the TM4SF19 promoter region, the prediction tool Ciiider also identified SREBF1 and SREBF2 binding sites. In the context of lipid processing macrophages in adipose tissue, this might be of particular interest, as this transcription factor is well known to be activated by sterols and regulate lipid metabolism. Did the authors also follow up on this? It would be an easy experiment to stimulate macrophages with cholesterol instead of LPS and perform CHIP. I agree that the provided data supports the conclusion of an obesity-induced pro-inflammatory trigger causing TM4SF19 expression. However, upon accumulation of lipids such as it occurs in obesity, this route of TM4SF19 induction seems similarly important to me.
2. Based on single cell transcriptomic data, the authors describe in Figure 4 and Figure 5 (and related supplementary figures) that upon HFD in gWAT mainly two subsets of Mac develop: Mac.Lyve1 and Mac Trem2. They further show that in obese gWAT less Mac Trem2 but more Mac.Lyve1 develop upon loss of Tm4sf19. In addition, based on presence of specific genes, they refine the roles of these obesity-induced macrophage

subsets and show, that they carry out distinct roles namely lipid handling and lysosomal activation (Trem2+, Mod 4 enriched Macs) as well as phagocytosis and anti-inflammatory restoration (Lyve1, Mod 2 enriched Macs). Following these findings, the authors perform an in-vitro culture experiment (RAW264.7 macrophages and dying adipocytes) monitoring levels of TREM2 and LYVE1 during lipid clearance to verify their sequencing data. First, for this in vitro study, the authors show protein levels after 8 hours of co-culture and gene expression data after 24 and 72 hours. Why do they not give a complete picture: protein levels at 8, 24 and 72 h) and/or gene expression at 8, 24 and 72h)?

Next, while authors verify the existence of these two cell populations and the effects of Tm4sf19 ko by flow analysis (Fig 5 F, G), they do not verify that the identified subpopulations really fulfil suggested function. I would recommend to isolate these cells from gWAT of HFD fed mice and perform ex vivo analysis (e.g. assess the phagocytic capacity of MacLyve1 Mod2 vs Trem2+ Mod 4 cells). Especially presenting C1qa, C1qb and C1qc as phagocytosis markers without prove of impacted phagocytosis may not be sufficient, as these markers might rather effect coagulation.

3. The conclusion, that “resolution and clearance of dying/dead adipocytes by macrophages is increased in TM4SF19 KO mice.” (line 300) must not be drawn. While Figure 5J and Supplemental video 2 provide in vivo evidence for phagocytosis in Tm4sf19 KO mice using an elegant model, data from the corresponding WT experiment is not shown. In order to support the conclusion, presentation of the WT data is crucial.

4. Line 314-316: “These data indicate that efficient efferocytosis of Tm4sf19 KO macrophages mitigates the pro-inflammatory environment induced by HFD, promotes tissue expansion by adipocyte hyperplasia, and restores mitochondrial metabolism and insulin sensitivity within the adipocytes of obese adipose tissue.” I did not see data regarding mitochondrial metabolism. The authors must not state this without supporting evidence.

5. The authors nicely show that HFD-induced impaired insulin-sensitivity and glucose tolerance is partly restored in Tm4sf19 KO mice. They provide data that adipocytes in HFD-fed Tm4sf19 KO mice are rather hyperplastic as the increase in number. Using single cell transcriptomic data, they describe the development of different adipocyte sub clusters during HFD feeding. They claim, that an insulin sensitive cluster (adipocyte 1) is being replaced by a rather lipid storing adipocyte cluster (adipocyte 3) upon HFD and that this progression is partly inhibited upon loss of Tm4sf19. They suggest that “ the expansion of

Adipocyte.1 in TM4SF19 KO restored the insulin sensitivity of adipocytes during high-fat feedings” or that “ that Tm4sf19 KO reverses insulin resistance in adipocytes” (line 429-430). As far as I understand, the authors thus suggest a cell autonomous effect explaining the increased insulin sensitivity in Tm4sf19 KO mice. As both, global and macrophage specific Tm4sf19 KO and also WT receiving BM from Tm4sf19 KO mice show dramatically reduced body weight gain, I wonder if the insulin sensitizing effect is secondary to the reduced weight gain, rather than to the existence of hyperplastic adipocytes. In order to verify a cell autonomous effects, the authors might assess insulin sensitivity and lipolytic capacity of primary adipocytes isolated from WT or Tm4sf19 KO mice.

6. In the same line, the observed reduction in CLS upon loss of Tm4sf19 might similarly be a secondary effect of reduced body weight gain. How do the authors explain the dramatic differences in body weight mechanistically?

7. Related to Figure 7: I wonder whether body composition and body weight was measured at the same time point. How is it possible, that bw is almost 60g in HFD WT mice, but lean and fat mass add up to only 45g?

8. How did the authors perform energy expenditure analysis? The method section does not contain any details regarding the statistical analysis. Since the figure legends of VO₂, VCO₂ and EE contain a kg, the data must be normalized somehow in Figure 7 H. To me, this seems to be normalized to body weight. However, if normalized to a higher body weight, energy expenditure might be artificially reduced and thus underestimated. Also in Figure S10 data is normalized to lean mass. Please refer to PMID: 22205519 in order to accurately present data on indirect calorimetry. In this line, I wonder if the energy expenditure is indeed enhanced upon loss of Tm4sf19 loss and if this might really explain the reduced body weight gain and improved metabolic parameters. First, in order to provide more evidence to support this hypothesis, I suggest the authors exclude a reduction in nutrient absorption potentially caused by loss of Tm4sf19. Did the authors study potential effects on intestinal lipid absorption?

9. Line 436-439: The authors discuss: “Consequently, the enhanced phagocytic activity of TREM2+ macrophages by TM4SF19 KO facilitates restorative adipose tissue remodeling, leading to increased systemic energy expenditure and improved insulin sensitivity.” First, the authors do not show, that TRem2+ mac from TM4SF19 KO mice have an enhanced phagocytic activity and next, especially trem2 + Mac are strongly reduced in TM4SF19 KO

mice so that I wonder how they can mediate adipose tissue remodeling.

Are there any effects of the TM4SF19 KO on macrophage immune function in other tissues than adipose?

Minor:

Fig2c: Which band was quantified for the membrane fraction of ATP6V0B? By eye, it looks like the band is more intense versus the WT, however, the quantification shows no difference.

FigS2: is not introduced correctly. In the text it says "TM4SF19 OE decreased lysosomal V-ATPase activity in RAW264.7 cells, while its KO in BMDMs increased (Fig. 2E and Fig. S2)." However the blots in Fig S2 only confirm the purity of lysosomal membrane fractions. Please correct.

Can you describe the phagocytosis study presented in Figure 2D in more detail? For instance, at what ratios were the adipocytes added to the macrophages? Compared to the in vivo situation where presumably adipocytes outnumber macrophages, it seems that in this setting, macrophages outnumber adipocytes.

With regards to the sentence in line 209: "Tm4sf19 was selectively expressed in Mac.Trem2 subtype (Fig. 3F and Fig. S4C)" I would suggest to change it into Tm4sf19 was mainly expressed in Mac.Trem2 subtype as in WT HFD, there is, albeit low, expression in monocytes and LEC.

Figure S4H should be introduced later as it also appears later in the text

I would suggest to transform Figure S5F-I into Suppl. Figure 7 as this fits better with the appearance in the text. Subsequently the other should be adjusted.

Line 385: "from GWAT (Fig. 7K)." should be "from GWAT (Fig. 7K-L).

Reviewer #2 (Remarks to the Author):

This work has good experimental design and rigorous proof of concept. The authors validate that TM4SF19 is up-regulated in TREM2+ macrophages that are recruited to adipose tissue during high-fat feeding. Over-expression of TM4SF19 could inhibit the acidification of lysosome, impede the lipid clearance, and suppress the efferocytosis in adipose tissues. Knockout of TM4SF19 in macrophage enhanced the adipogenesis and limits the progression

of diabetes in high-fat diet mouse model. These evidences support author's hypothesis that enhancing the efficiency of lipid handling and lysosomal processing could accelerate clearance and resolution by adipose tissue macrophages. I suggest to accept this manuscript with minor revisions suggested below:

1. According to an article published on Autophagy 2019, VOL. 15, NO. 6, 960–975, ATP6V0D2 is a macrophage-specific sub-unit of V-ATPase that facilitate autophagosome-lysosome fusion and restrict inflammasome activation. Author also observe the expression of this gene in Mod4, where Tm4sf19 is included (Fig. 4c). This indicates that ATP6V0D2 might be also a target of Tm4sf19 and author better investigate it in IP study. If positive, then the western blot in BMDM of KO mice and RAW cells should be evaluated too.
2. Fig. 5D only shows a 500 x 500 um area. To avoid selection bias due to regional variation, for each tissue slice (biological replica), it's better to randomly select 6-10 area (technical replica) for accurate statistics and fair comparison.
3. Please show GWAT histopathology of wild-type & TM4SF19 KO mice corresponding to Fig. S7.

Reviewer #3 (Remarks to the Author):

In the manuscript titled “TM4SF19-mediated control of lysosomal activity in macrophages contributes to obesity-induced inflammation and metabolic dysfunction,” Choi et al. offer a comprehensive study of TM4SF19's function in adipose tissue macrophages and its implication in obesity-related metabolic impairment. The paper leverages an array of methodologies, including gene module analysis, single-nucleus RNA sequencing (snRNA-seq), conditional knockout models in mice, and diverse imaging techniques to decipher TM4SF19's role. The authors suggest that TM4SF19 ablation in macrophages may safeguard against high-fat diet (HFD)-induced obesity and enhance metabolic processes, providing significant insights into the relationship between TM4SF19, macrophage activity, and inflammation regulation within the obesity framework.

While the manuscript's strengths are evident, it could be enhanced by addressing the

following points:

1. The study's reliance on mouse models, although yielding promising outcomes, leaves the translational potential to human physiology uncertain. Questions about the conservation of effects in human macrophages and adipose tissue within the milieu of metabolic disorders need addressing to bridge this gap.
2. The identification of TM4SF19 as a modulator of lysosomal V-ATPase activity is pivotal, yet the specific molecular pathways through which TM4SF19 impacts macrophage functionality and systemic metabolism remain partially unresolved. An expansion of experimental design to delineate these pathways would enrich the understanding of these intricate interactions.
3. In Figure 2, the authors revealed that TMS4F19 is located in lysosome membrane. Knockout of TMS4F19 in BMDM results in a reduction of lysosomal pH, which suggests an improvement in lysosome function. However, in Figure 4C, TMS4F19+ Mod4 Mac population exhibited lower Vacuolar acidification, as evidenced by the expression of ATP6V1H1 and other genes in KO HFD group. Does this contradict the observations in Figure 2F?

For minor issues:

1. In Figure 7G, it would be beneficial to include the raw glucose level data to maintain consistency with Figure 7F and enhance the interpretability of the insulin tolerance test (ITT) results.
2. The use of 'scRNAseq' should be rectified to 'snRNAseq' throughout the manuscript to reflect the techniques accurately used in the study.

Addressing these concerns would greatly solidify the manuscript's contributions to the field and provide a clearer picture of the potential clinical relevance of the findings.

We thank the reviewers for their valuable comments. We have undertaken revisions, incorporating new experimental data to address the concerns raised during the review.

The major revisions include:

1. Two-photon intravital imaging capturing an adipocyte surrounded by macrophages (crown-like structure) in GWAT of WT mice (**Figure 5J and Video S2**).
2. Reanalysis of energy expenditure data using regression-based analysis of covariance (ANCOVA) with body mass as a covariate (**Figure 7A-B, 8H-I, and S3F-G**).
3. IP analysis demonstrating the interaction between TM4SF19 and ATP6V0D2 (**Figure 2B**).
4. ChIP analysis demonstrating the recruitment of SREBP1 to the Tm4sf19 promoter in RAW264.7 cells (**Figure S1C-D**).
6. qPCR analysis of *Trem2*, *Lyve1*, *Tnf*, and *Arg1* in RAW264.7 cells co-cultured with dead/dying adipocytes (**Figure S6C**).
7. Assessment of lipid content in feces of WT, Tm4sf19 global KO, and macrophage-specific Tm4sf19 KO mice under chow or high-fat diet feeding (**Figures S3H, S10B, and S11C**).
8. Immunoblot analysis of insulin signaling and lipolysis in adipocytes differentiated from precursor cells in GWAT of TM4SF19 KO and WT mice.

We specified the use of male mice in the abstract and method sections. We also provide information on the sex/gender of animal models and human samples in the manuscript, as detailed in the Reporting Summary. Additionally, we have included source data.

REVIEWER COMMENTS

Reviewer #1 (Remarks to the Author):

Choi et al identify enhanced expression of Tm4sf/TM4SF19 in obese adipose tissue of mice and man. They identify TM4SF19 to be expressed by macrophages strongly induced upon HFD. Using elegant in vitro studies, they identify TM4SF19 as an interaction partner of ATP6V0B preventing the assembly of a lysosomal V-ATPase and thus regulating lysosomal acidification. They show that in BMDM loss of TM4SF19 reduced lysosomal acidification and enhanced the clearance of dying adipocytes. In vivo, they find that loss of TM4SF19 shifts macrophage and adipocyte dynamics overall resulting in reduced body weight gain, enhanced glucose and insulin sensitivity and reduced inflammatory responses.

The paper is of particular interest as it addresses the fine-tuning of tissue remodeling processes by macrophages in the context of obesity, which might contribute to identifying potential therapeutic targets for the treatment of obesity and associated disorders. Using elegant in vitro and in vivo models as well as state of the art sequencing technologies the authors identify a novel regulator of macrophage biology and uncover a strong phenotyp depending on the lysosomal

protein TM4SF19. However, I have some remarks which need to be addressed by the authors:

1. Besides the characterized NFkB binding side in the TM4SF19 promoter region, the prediction tool Ciider also identified SREBF1 and SREBF2 binding site. In the context of lipid processing macrophages in adipose tissue, this might be of particular interest, as this transcription factor is well known to be activated by sterols and regulate lipid metabolism. Did the authors also follow up on this? It would be an easy experiment to stimulate macrophages with cholesterol instead of LPS and perform CHIP. I agree that the provided data supports the conclusion of an obesity-induced pro-inflammatory trigger causing TM4SF19 expression. However, upon accumulation of lipids such it occurs in obesity, this route of TM4SF19 induction seems similarly important to me.

Response: We thank the reviewer for the insightful suggestion. In this revision, RAW 264.7 cells were treated with **oxLDL (100 ug/ml)** and subsequent ChIP analysis was conducted. We found oxLDL treatment upregulated SREBP1 levels and the enrichment of SREBP1 binding to the promoter region of *Tm4sf19*. The data are included as supplemental **Figure S1C-D**

2. Based on single cell transcriptomic data, they authors describe in Figure 4 and Figure 5 (and related supplementary figures) that upon HFD in gWAT mainly two subsets of Mac develop : Mac.Lyve1 and Mac Trem2. They further show that in obese gWAT less Mac.Trem2 but more Mac.Lyve1 develop upon loss of Tm4sf19. In addition, based on presence of specific genes, they refine the roles of these obesity-induced macrophage subsets and show, that they carry out distinct roles namely lipid handling and lysosomal activation (Trem2+, Mod 4 enriched Macs) as well as phagocytosis and anti-inflammatory restoration (Lyve1, Mod 2 enriched Macs). Following these finding, the authors perform an in-vitro culture experiment (RAW264.7 macrophages and dying adipocytes) monitoring levels of TREM2 and LYVE1 during lipid clearance to verify their sequencing data. First, for this in vitro study, the authors show protein levels after 8 hours of co-culture and gene expression data after 24 and 72 hours. Why do they not give a complete picture: protein levels at 8, 24 and 72 h) and/or gene expression at 8 , 24 and 72h)?

Response: In response to the reviewer's recommendation, we conducted qPCR analysis of macrophages at 0, 8, 24, and 72 hours. Consistent with our previous findings, *Trem2* and *Tnf* expression peaked at 8 hours and gradually declined at 24 and 72 hours, while *Lyve1* and *Arg1* expression exhibited a temporal decrease at 8 hours, followed by a progressive increase over the course of 72 hours. The data are provided as supplemental **Figure S6C**

Next, while authors verify the existence of these two cell populations and the effects of Tm4sf19 ko by flow analysis (Fig 5 F, G), they do not verify that the identified subpopulations really fulfil suggested function. I would recommend to isolate these cells from gWAT of HFD fed mice and perform ex vivo analysis (e.g. assess the phagocytic capacity of MacLyve1 Mod2 vs Trem2+ Mod4 cells). Especially presenting C1qa, C1qb and C1qc as phagocytosis markers without prove of impacted phagocytosis may not be sufficient, as these markers might rather effect coagulation.

Response: In our efforts to compare the phagocytic capacity of Lyve1+Mod2 and Trem2+ Mod4 cells, we encountered technical challenges in isolating macrophages with precisely matched molecular characteristics for each population. Attempting to sort cells based on single surface markers, Trem2 and Lyve1, did not precisely match the mod2 and mod4 cells, introducing complexity to the analysis.

Moreover, in vivo phagocytosis likely involves signals that are highly localized in the microenvironment. Thus, during dissociation and flow sorting, cells would lose microenvironmental signaling that could be necessary for phagocytic activity,

To address the reviewer's concern, in this revised manuscript, we have tempered our interpretations by removing the assertion that C1qa and C1qb are definitive phagocytosis markers. In this revised manuscript, we repeated our analysis using *Mertk* (*Mer*)¹, *Cd163*², and *Stab1*³ instead of C1qa, C1qb, and C1qc as phagocytosis markers, as shown in Figure 4C. These genes are part of Mod2 genes and exhibit similar expression patterns to C1qa, C1qb, and C1qc. Additionally, we have also made corresponding revisions to the figures related to phagocytosis with these markers (Figure 4D,G). In our revised discussion, we openly acknowledge the limitations of our study and emphasize the need for future investigation to thoroughly explore the phagocytic activity of these cells. In the discussion: "While we considered *Mertk*, *Cd163* and *Stab1* as potential phagocytosis markers in our snRNA-seq data analysis, further studies are needed to conclusively establish the phagocytic activity of macrophage subtypes identified in the current study."

G. Module scores for each biological term in the macrophage population are shown in a PCA plot.

3. The conclusion, that “resolution and clearance of dying/dead adipocytes by macrophages is increased in TM4SF19 KO mice.” (line 300) must not be drawn. While Figure 5J and Supplemental video 2 provide in vivo evidence for phagocytosis in Tm4sf19 KO mice using an elegant model, data from the corresponding WT experiment is not shown. In order to support the conclusion, presentation of the WT data is crucial.

Response: In response to the reviewer's comments, we incorporated intravital imaging of the corresponding condition in WT mice. While we identified crown-like structure zones (clusters of CX3CR1-GFP+ cells), there was an absence of phagocytosis activity in the cells within the same time frame observed in the KO mice (Revised Figure 5J and supplemental Video 2).

4. Line 314-316: “These data indicate that efficient efferocytosis of Tm4sf19 KO macrophages mitigates the pro-inflammatory environment induced by HFD, promotes tissue expansion by adipocyte hyperplasia, and restores mitochondrial metabolism and insulin sensitivity within the adipocytes of obese adipose tissue.” I did not see data regarding mitochondrial metabolism. The authors must not state this without supporting evidence.

Response: We assessed mitochondrial content, as supported by supplemental figure S9E. Due to its significance recognized by the reviewer, we have it as a main Figure 7 in this revised manuscript.

5. The authors nicely show that HFD-induced impaired insulin-sensitivity and glucose tolerance is partly restored in *Tm4sf19* KO mice. They provide data that adipocytes in HFD-fed *Tm4sf19* KO mice are rather hyperplastic as the increase in number. Using single cell transcriptomic data, they describe the development of different adipocyte sub clusters during HFD feeding. They claim, that an insulin sensitive cluster (adipocyte 1) is being replaced by a rather lipid storing adipocyte cluster (adipocyte 3) upon HFD and that this progression is partly inhibited upon loss of *Tm4sf19*.

They suggest that “ the expansion of Adipocyte.1 in TM4SF19 KO restored the insulin sensitivity of adipocytes during high-fat feedings” or that “ that Tm4sf19 KO reverses insulin resistance in adipocytes” (line 429-430). As far as I understand, the authors thus suggest a cell autonomous effect explaining the increased insulin sensitivity in Tm4sf19 KO mice. As both, global and macrophage specific Tm4sf19 KO and also WT receiving BM from Tm4sf19 KO mice show dramatically reduced body weight gain, I wonder if the insulin sensitizing effect is secondary to the reduced weight gain, rather than to the existence of hyperplastic adipocytes. In order to verify a cell autonomous effects, the authors might assess insulin sensitivity and lipolytic capacity of primary adipocytes isolated from WT or Tm4sf19 KO mice.

Response

As adipocytes do not express detectable levels TM4SF19 protein and macrophage-specific Tm4sf19 KO phenocopies whole-body KO of Tm4sf19, we concluded that the effects of Tm4sf19 KO on adipocyte function must be secondary to inactivation in macrophages. We acknowledge that multiple pathways downstream of macrophage function likely contribute to the improvement in systemic insulin sensitivity, including reduced inflammatory cytokine signaling and mitochondrial metabolism.

In addressing the reviewer’s concerns, we isolated stromal cells from GWAT of TM4SF19 KO and WT mice and subsequently differentiated them into adipocytes. Due to the macrophage-specific expression of Tm4sf19, the TM4SF19 protein was not detectable in primary adipocytes from either WT or KO mice, as confirmed by western blot analysis. Upon acute insulin treatment for 15 mins, the increase in insulin downstream signaling (p-AKT/AKT and p-IRS/IRS1) was induced in both KO and WT. Isoproterenol treatment led to similar increases in p-HSL/HSL levels in both groups, and no significant difference was observed between WT and KO. Furthermore, the levels of FFA and glycerol in the medium following isoproterenol treatment were not affected by TM4SF19 KO. These data suggest that cell-autonomous effects do not significantly contribute to the phenotype adipocytes in TM4SF19 KO mice (insulin sensitivity and lipolytic capacity), supporting our initial hypothesis

6. In the same line, the observed reduction in CLS upon loss of Tm4sf19 might similarly be a secondary effect of reduced body weight gain. How do the authors explain the dramatic differences in body weight mechanistically?

Response:

We appreciate the insightful comments from the reviewer.

Tm4sf19 is highly enriched in adipose tissue macrophages, which that the major source of proinflammatory signaling in adipose tissue during high fat feeding. To test the direction of

causality, we deleted *Tm4sf19* in macrophages and found that macrophage-specific KO phenocopies the effect of whole-body KO. Thus, we conclude that changes in body weight are a consequence of primary effects of *Tm4sf19* KO in macrophages. In parallel, we show that macrophage-specific KO accelerates CLS clearance *in vivo* and *in vitro*, promotes new fat cell formation, and increases mitochondrial oxidative gene expression, each of which likely contributes to elevated energy expenditure and reduced overall adiposity. Certainly, reduction in fat accumulation, particularly in hypertrophic adipocytes, could further reduce inflammation and promote energy expenditure in a complex feedback relationship that begins with *Tm4sf19* function in macrophages.

7. Related to Figure 7: I wonder whether body composition and body weight was measured at the same time point. How is it possible, that bw is almost 60g in HFD WT mice, but lean and fat mass add up to only 45g?

Response: We apologize for any confusion. As pointed out by the reviewer, body composition (lean and fat mass) and indirect calorimetry in the original submitted data were initially measured at 5 weeks of HFD feeding, which falls in the middle of the body weight monitoring period. In this revised version, we have included new data on body composition and indirect calorimetry data at 12 weeks of HFD feeding, aligning with the end point of body weight monitoring. The text and figure legends explicitly indicate when the analysis was conducted. Notably, the data at 5 weeks and 12 weeks displayed a similar pattern, with reduced body fat contents observed in *Tm4sf19* KO. Individual values including body weight and body composition can be found in the Source Data.

8. How did the authors perform energy expenditure analysis? The method section does not contain any details regarding the statistical analysis. Since the figure legends of VO_2 , VCO_2 and EE contain a kg, the data must be normalized somehow in Figure 7 H. To me, this seems to be normalized to body weight. However, if normalized to a higher body weight, energy expenditure might be artificially reduced and thus underestimated. Also in Figure S10 data is normalized to lean mass. Please refer to PMID: 22205519 in order to accurately present data on indirect

calorimetry. In this line, I wonder if the energy expenditure is indeed enhanced upon loss of Tm4sf19 and if this might really explain the reduced body weight gain and improved metabolic parameters. First, in order to provide more evidence to support this hypothesis, I suggest the authors exclude a reduction in nutrient absorption potentially caused by loss of Tm4sf19. Did the authors study potential effects on intestinal lipid absorption?

Response: We thank the reviewer for the constructive comment. We agree that the methods suggested are a more accurate measurement of energy expenditure in mice that differ in body mass. In response to the suggestion, we reanalyzed energy expenditure data using regression-based analysis of covariance (ANCOVA) with body mass as a covariate (PMID: 22205519). The results showed a significant effect of whole-body and macrophage-specific Tm4sf19 KO on energy metabolism when we compared ANCOVA-adjusted values (revised Figure 8H and I). Consistently, we reported enhanced mitochondrial contents and metabolism in adipose tissue of Tm4sf19 KO in the original manuscript.

Additionally, as the reviewer suggested, we tested intestinal lipid absorption using a modified Folch extraction method⁴. There was no significant difference between WT and KO mice in lipids extracted from 1 g of feces. (**Fig.S3H, S10B, and S11C**)

Figure S11. Indirect calorimetry analysis of TM4SF19 MKO and WT mice, related to Figure 8. A-B. Indirect calorimetry analysis of WT and macrophage-specific TM4SF19 MKO mice after 5 weeks (A) and 12 weeks (B) of HFD feeding (n = 6). VO₂, VCO₂, and energy expenditure normalized by body weight (n = 6). C. Lipid content in feces (n = 4).

9. Line 436-439: The authors discuss: “Consequently, the enhanced phagocytic activity of TREM2⁺ macrophages by TM4SF19 KO facilitates restorative adipose tissue remodeling, leading to increased systemic energy expenditure and improved insulin sensitivity.” First, the authors do not show, that TREM2⁺ mac from TM4SF19 KO mice have an enhanced phagocytic activity and next, especially trem2 + Mac are strongly reduced in TM4SF19 KO mice so that I wonder how they can mediate adipose tissue remodeling.

Response: We acknowledge that the sentence highlighted by the reviewer may be somewhat misleading, and we apologize for any confusion. To clarify the intended meaning of the sentences, we removed "Trem2+" from the phrase. We conclusively demonstrated that the phagocytic activity of macrophages is enhanced by Tm4sf19 KO, by both in vivo intravital imaging and in vitro experiments in our study.

Are there any effects of the TM4SF19 KO on macrophage immune function in other tissues than adipose?

Response: We appreciate the reviewer's comments. We note in the revised discussion that TREM2⁺ cells are observed in a variety of pathological conditions, such as hepatosteatosis and atherosclerosis, and suggest that our results provide a foundation for investigating the role of TM4SF19 in these disease conditions. We have planned such experiments; however, we hope that the reviewer will agree that these studies are beyond the scope of the current report.

Minor:

Fig2c: Which band was quantified for the membrane fraction of ATP6V0B? By eye, it looks like the band is more intense versus the WT, however, the quantification shows no difference.

Response: We apologize for any confusion. We identified the protein ATP6V0B based on the expected protein size (~21 kd), and the quantified bands are marked with arrows in the revised manuscript. The quantification of ATP6V0B in the membrane fraction (Figure 2C) showed a ratio of 1:1.13 (WT: KO), while ATP6V1B2 exhibited a ratio of 1:1.44 (WT: KO). This indicates that the knockout (KO) increased the association of the V1 complex (ATP6V1B2) with the membrane compartment when normalized to the V0 complex component (ATP6V0B)."

The bar graph represents the average values of ATP6V0B/LAMP1 from six replicates, with no significant difference observed between WT and KO. Individual values are provided in the Source Data, and all images of immunoblot data are presented below.

FigS2: is not introduced correctly. In the text it says “TM4SF19 OE decreased lysosomal V-ATPase activity in RAW264.7 cells, while its KO in BMDMs increased (Fig. 2E and Fig. S2).” However the blots in Fig S2 only confirm the purity of lysosomal membrane fractions. Please correct.

Response: We apologize for any confusion. We removed “Fig.S2’ from the end of the sentence (line 181) and appropriately added it earlier in the text (line 179).

“we isolated lysosomal fractions from macrophages and measured V-ATPase activity in lysosomal fractions (**Fig. S2**). As expected, TM4SF19 OE decreased lysosomal V-ATPase activity in RAW264.7 cells, while its KO in BMDMs increased (**Fig. 2E**).”

Can you describe the phagocytosis study presented in Figure 2D in more detail? For instance, at what ratios were the adipocytes added to the macrophages? Compared to the in vivo situation where presumably adipocytes outnumber macrophages, it seems that in this setting, macrophages outnumber adipocytes.

Response: In this revised manuscript, we provided additional details about the experimental procedures in the Methods section (line 924), “BODIPY-labeled C3H10T1/2 cells were detached, added to the DiO- labeled macrophages in a 1:1 ratio.”

As noted by the reviewer, we acknowledge that this condition does not precisely replicate the in vivo situation. Under basal conditions (during chow diet feeding), the number of adipocytes exceeds that of macrophages.

However, after 8 weeks of HFD feeding, the macrophage population outnumbers adipocytes in WT and is similar in TM4SF19 KO mice, based on our snRNAseq data and others' reports. For experimental feasibility, we employed a 1:1 ratio of adipocytes to macrophages.

Additionally, it is crucial to recognize that in the in vivo condition, specifically focusing on crown-like structures, one adipocyte is often surrounded by a large number of macrophages, as they are specifically recruited to dying/dead adipocytes in the adipose tissue of high-fat diet-induced obese mice. Thus, after HFD feeding, especially in this local area, it is not the case that adipocytes outnumber macrophages.

With regards to the sentence in line 209: “Tm4sf19 was selectively expressed in Mac.Trem2 subtype (Fig. 3F and Fig. S4C)” I would suggest to change it into Tm4sf19 was mainly expressed in Mac.Trem2 subtype as in WT HFD, there is, albeit low, expression in monocytes and LEC.

Response: According to the reviewer's suggestion, we revised the sentences in lines 213-214.

Figure S4H should be introduced later as it also appears later in the text

I would suggest to transform Figure S5F-I into Suppl. Figure 7 as this fits better with the appearance in the text. Subsequently the other should be adjusted.

Response: We appreciate the reviewer for bringing the figure order errors to our attention. To address this, we have repositioned **Figure S4H-I** to **Figure S8** in accordance with the order in the text. Additionally, we have followed the reviewer's suggestion and reorganized **Figure S5F-I** into **Figure S7**.

Line 385: “from GWAT (Fig. 7K).” should be “from GWAT (Fig. 7K-L).

Response: We apologize for any confusion. We revised our sentences in revised manuscript in line 392 (revised Figure **8L-M**).

Reviewer #2 (Remarks to the Author):

This work has good experimental design and rigorous proof of concept. The authors validate that TM4SF19 is up-regulated in TREM2⁺ macrophages that are recruited to adipose tissue during high-fat feeding. Over-expression of TM4SF19 could inhibit the acidification of lysosome, impede the lipid clearance, and suppress the efferocytosis in adipose tissues. Knockout of TM4SF19 in macrophage enhanced the adipogenesis and limits the progression of diabetes in high-fat diet mouse model. These evidences support author's hypothesis that enhancing the efficiency of lipid handling and lysosomal processing could accelerate clearance and resolution by adipose tissue macrophages. I suggest to accept this manuscript with minor revisions suggested below:

1. According to an article published on Autophagy 2019, VOL. 15, NO. 6, 960–975, ATP6V0D2 is a macrophage-specific sub-unit of V-ATPase that facilitate autophagosome-lysosome fusion and restrict inflammasome activation. Author also observe the expression of this gene in Mod4, where Tm4sf19 is included (Fig. 4c). This indicates that ATP6V0D2 might be also a target of Tm4sf19 and author better investigate it in IP study. If positive, then the western blot in BMDM of KO mice and RAW cells should be evaluated too.

Response: We appreciated the reviewer's insightful comments. Following the reviewer's recommendation, we detected ATP6V0D2 in immunoprecipitated fractions and demonstrated the interaction between TM4SF19 and ATP6V0D2.

2. Fig. 5D only shows a 500 x 500 um area. To avoid selection bias due to regional variation, for each tissue slice (biological replica), it's better to randomly select 6-10 area (technical replica) for accurate statistics and fair comparison.

Response: Following the reviewer's suggestion, we performed a reanalysis of the data using six biological replicates. Specifically, for each biological replicate of each condition (a total of six WT and six KO mice), we randomly selected six areas for each tissue section as technical replica.

The data are presented as average F4/80 staining intensity values from each condition. We have outlined these approaches in the Method section for clarity

3. Please show GWAT histopathology of wild-type & TM4SF19 KO mice corresponding to Fig. S7.

Response: According to the reviewer's comments, we stained paraffin sections of GWAT of WT and TM4SF19 KO PDGFRA-Cre/LSL-tdTomato/Cx3Cr1-GFP mice with hematoxylin and eosin (H&E).

Reviewer #3 (Remarks to the Author):

In the manuscript titled “TM4SF19-mediated control of lysosomal activity in macrophages contributes to obesity-induced inflammation and metabolic dysfunction,” Choi et al. offer a comprehensive study of TM4SF19’s function in adipose tissue macrophages and its implication in obesity-related metabolic impairment. The paper leverages an array of methodologies, including gene module analysis, single-nucleus RNA sequencing (snRNA-seq), conditional knockout models in mice, and diverse imaging techniques to decipher TM4SF19’s role. The authors suggest that TM4SF19 ablation in macrophages may safeguard against high-fat diet (HFD)-induced obesity and enhance metabolic processes, providing significant insights into the relationship between TM4SF19, macrophage activity, and inflammation regulation within the obesity framework.

While the manuscript’s strengths are evident, it could be enhanced by addressing the following points:

1. The study’s reliance on mouse models, although yielding promising outcomes, leaves the translational potential to human physiology uncertain. Questions about the conservation of effects in human macrophages and adipose tissue within the milieu of metabolic disorders need addressing to bridge this gap.

Response: Despite transcriptional differences observed between adipose tissue macrophages in human and mice, we have some clues suggesting a potential similar function of TM4SF19 in human adipose tissue. Notably, we identified a positive correlation between TM4SF19 and BMI in human adipose tissue (Figure 1J). To further investigate its role, we analyzed public sn/scRNA-seq data⁵, revealing TM4SF19 is expressed in the TREM2-expressing macrophage subset (hMac2, as specified in the paper) and this macrophage subset has higher lipid handling and vacuolar acidification module scores compared to the other macrophage subset (hMac1 and hMac3) (Figure S5E).

2. The identification of TM4SF19 as a modulator of lysosomal V-ATPase activity is pivotal, yet the specific molecular pathways through which TM4SF19 impacts macrophage functionality and systemic metabolism remain partially unresolved. An expansion of experimental design to delineate these pathways would enrich the understanding of these intricate interactions.

Response: We agree that our observation that macrophage Tm4sf19 is a major regulator of diet-induced proinflammatory signaling in adipose tissue is a foundational discovery. Mechanistically, we demonstrate that Tm4sf19 is upregulated by lipid-sensing and proinflammatory transcription factors. Furthermore, we demonstrate that Tm4sf19 is a lysosomal protein and key regulator of lysosomal acidification through direct interactions with v-ATPase. Our work focused on established proximal effects of lysosomal function in adipose tissue, namely, clearance of defective adipocytes and adipocyte hyperplasia. We agree that improved macrophage function likely affects several parallel pathways through secondary signaling. In this regard, our comprehensive experimental design provides extensive sc- and sn-RNA-seq data sets, as well as in-depth metabolic phenotyping, which provide a foundation for followup experimentation. Our revised discussion acknowledges that the full array of secondary effects of Tm4sf19 KO remain to be explored. Nonetheless, our results show that targeting this key, diet-induced regulator has broad beneficial metabolic effects, and strongly suggests that Tm4sf19 could be a novel point of therapeutic intervention for obesity-related diseases.

3. In Figure 2, the authors revealed that TMS4F19 is located in lysosome membrane. Knockout of TMS4F19 in BMDM results in a reduction of lysosomal pH, which suggests an improvement in lysosome function. However, in Figure 4C, TMS4F19+ Mod4 Mac population exhibited lower Vacuolar acidification, as evidenced by the expression of ATP6V1H1 and other genes in KO HFD group. Does this contradict the observations in Figure 2F?

Response: We apologize to the reviewer for the confusion regarding the expression of genes involved in vacuolar acidification. We would like to emphasize that the experiments in Figure 2 were conducted to elucidate the mechanism of TM4SF19, and there was no change in the total amount of ATP6V0B protein with KO or OE of TM4SF19 in BMDM and RAW264.7 cells (Figure 2C and 2D). In Figure 4C, we propose that the reduced expression of vacuolar

acidification-related genes in HFD KO mice compared to WT HFD condition reflects a less pathological state.

To elaborate, we hypothesize that under conditions of over-nutrition, such as with HFD, TM4SF19 acts as an inhibitor of V-ATPase. This reduces lysosomal acidification and subsequently impedes the degradation of extracellular materials and dead adipocytes. However, during HFD feeding, there is an increased need for lysosomal activity to remove debris and lipids from dead/dying adipocytes in adipose tissue. Consequently, as shown in Figure 4C, the expression of genes related to vacuolar acidification is elevated in HFD-fed mice compared to NCD-fed conditions.

Deletion of TM4SF19 under HFD-fed condition enhances lysosomal function, leading to a more rapid clearance of dying adipocytes (Figure 2E-G). Rapid efferocytosis reduces the magnitude and duration of inflammatory signals that recruit Trem2⁺ macrophages (Figure 5A-G), thereby reducing their accumulation and proinflammatory status. As a result, the overall pathological state of KO HFD mice is less severe than that of WT NCD mice, and the expression of genes associated with vacuolar acidification may not require such high levels. Therefore, we interpret that, at the transcriptomic level, the state of KO HFD mice becomes more similar to that of WT NCD mice, suggesting an attenuated pathological impact in the absence of TM4SF19.

Furthermore, we verified that there are no differences in the expression levels of *Atp6v1h*, *Atp6v0d2*, *Atp6v0a1*, *Atp6v1b2*, and *Atp6v1a* in BMDM obtained from *Tm4sf19* knockout mice. This finding further suggests that the removal of TM4SF19 does not directly affect the expression levels of vATPase genes. This is consistent with our findings indicating that TM4SF19 KO does not regulate vATPase gene transcript expression but functions by inhibiting vATPase complex assembly.

For minor issues:

1. In Figure 7G, it would be beneficial to include the raw glucose level data to maintain consistency with Figure 7F and enhance the interpretability of the insulin tolerance test (ITT) results.

Response: In response to the reviewer's suggestion, we have changed Figure 7G to include the raw glucose level data.

2. The use of 'scRNAseq' should be rectified to 'snRNAseq' throughout the manuscript to reflect the techniques accurately used in the study.

Addressing these concerns would greatly solidify the manuscript's contributions to the field and provide a clearer picture of the potential clinical relevance of the findings.

Response: Following the reviewer's suggestion, we corrected the term "scRNA-seq" to "snRNA-seq" throughout the manuscript to accurately reflect the techniques used in this study.

References used for the point-by-point responses

- 1 Dransfield, I., Zagorska, A., Lew, E. D., Michail, K. & Lemke, G. Mer receptor tyrosine kinase mediates both tethering and phagocytosis of apoptotic cells. *Cell death & disease* **6**, e1646, doi:10.1038/cddis.2015.18 (2015).
- 2 Schulz, D., Severin, Y., Zanotelli, V. R. T. & Bodenmiller, B. In-Depth Characterization of Monocyte-Derived Macrophages using a Mass Cytometry-Based Phagocytosis Assay. *Scientific reports* **9**, 1925, doi:10.1038/s41598-018-38127-9 (2019).
- 3 Park, S. Y., Bae, D. J., Kim, M. J., Piao, M. L. & Kim, I. S. Extracellular low pH modulates phosphatidylserine-dependent phagocytosis in macrophages by increasing stabilin-1 expression. *J Biol Chem* **287**, 11261-11271, doi:10.1074/jbc.M111.310953 (2012).
- 4 Kraus, D., Yang, Q. & Kahn, B. B. Lipid Extraction from Mouse Feces. *Bio Protoc* **5**, doi:10.21769/bioprotoc.1375 (2015).
- 5 Emont, M. P. *et al.* Author Correction: A single-cell atlas of human and mouse white adipose tissue. *Nature* **620**, E14, doi:10.1038/s41586-023-06445-2 (2023).

REVIEWERS' COMMENTS

Reviewer #1 (Remarks to the Author):

The authors have addressed my initial concerns carefully and clarified most of the raised points. I suggest the manuscript for publication after addressing some minor revisions. Additionally, I replied to the response of the authors as follows:

To 1:

I acknowledge the efforts of the authors and the interesting insights from following my suggestion. I think the presentation of the results as additional supplementary figure adds an important value on the physiological transcriptional regulation of TM4SF19.

To 2:

The authors now present a complete picture on the regulation of macrophage Trem2 and Lyve on gene expression and protein level during co-culture with adipocytes. I acknowledge that.

In addition, I understand the technical limitations the authors encountered during the isolation of the distinct cell populations for ex vivo phagocytosis assay and I acknowledge their attempts. With regards to my concerns of choosing C1qa, C1qb and C1qc as phagocytosis markers, I highly appreciate that the authors meanwhile considered Mertk, Cd163 as phagocytosis markers and updated the corresponding figures. In light with the additional point raised in the discussion, I think my concern is well addressed.

To 3: Adding the control group to Revised Figure 5J and supplemental Video 2 was essential in order to keep the conclusion that clearance of dying/dead adipocytes by macrophages is increased in TM4SF19 KO mice. I acknowledge the additional work of the authors.

To 4: I apologize to the authors for missing the data on mitochondrial metabolism in the previous version of the manuscript (Sup Fig 9.) I acknowledge, that the authors include these data in the new Figure 7, clearly supporting their conclusion. However, as the sentence in line 316-320 "These data indicate that efficient efferocytosis of Tm4sf19 KO macrophages mitigates the proinflammatory environment induced by HFD, promotes tissue expansion by adipocyte hyperplasia, and restores mitochondrial metabolism and insulin sensitivity within the adipocytes of obese adipose tissue." Still mentions the restoration of mitochondrial metabolism without any reference to Figure 7, I suggest to delete this part of the conclusion at this position in the manuscript and rather add it after the presentation and

description of Figure 7. On top of that, the same is true for insulin sensitivity. Also here, the conclusion is drawn before the actual data has been presented. In line with my previous suggestion, I again propose that the authors delete this part of the conclusion here.

To 5: I appreciate the clarification of this point and agree, that cell-autonomous effects do not significantly contribute to the phenotype of adipocytes in TM4SF19 KO mice and that rather potentially increased adipogenesis might be causal to the observed phenotype. Still, with regards to the TM4SF19 KO – mediated potential increase of de novo adipogenesis from progenitors, can the authors speculate, how this effect would be mediated. Along that line, perhaps it would be interesting to follow adipogenic differentiation of SVF fraction in the presence or absence of WT vs. vs TM4SF19 KO macrophages or their conditioned medium.

To 6: I thank the author for highlighting that the observed body weight differencers might be the result of a complex feedback relationship that begins with Tm4sf19 function in macrophages.

To 7: Thanks for adding the additional data and clearing this point.

To 8: I highly appreciate the effort of the authors to reanalyze data on energy expenditure using regression based analysis of covariance (ANCOVA) with body mass as a covariate as described in PMID: 22205519 as well as their efforts to investigate potential differences in intestinal lipid uptake and have no further concerns.

To 9: Deletion of Trem2+ indeed avoids conclusion.

To 10: I understand that this might go beyond the scope of this manuscript. However, if I am not mistaken, in the revised manuscript, the authors also present data generated from livers of WT and TM4SF19 KO- mice showing that the inflammatory response of HFD is indeed reduced in the KO similarly as observed in the WAT.

To Minor Fig2c: I appreciate the efforts of the authors.

To Minor FigS2: The point was addressed and I do not have any further concerns.

Thanks for giving more technical details on the phagocytosis assays. I understand the rational of the experiments and do not have any further concerns.

I acknowledge that all other minor points were sufficiently addressed.

Reviewer #2 (Remarks to the Author):

The author has addressed the issues and the manuscript is ready for publication.

Reviewer #3 (Remarks to the Author):

The authors have made substantial efforts, enhancing the clarity and comprehensibility of the manuscript significantly. This revision by Choi et al. has successfully addressed all the concerns I previously raised. The paper now stands out in terms of both quality and importance. Given its notable improvements and the relevance of its findings, I recommend its acceptance for publication in Nature Communications.

We thank the reviewers for their thorough review and constructive feedback during the revision process.

In response to Reviewer 1's comments, we removed one sentence from the manuscript as it may have conveyed assumptive expression before conclusive data presentation.

Point-by-point responses to the reviewers' comments

REVIEWERS' COMMENTS

Reviewer #1 (Remarks to the Author):

The authors have addressed my initial concerns carefully and clarified most of the raised points. I suggest the manuscript for publication after addressing some minor revisions. Additionally, I replied to the response of the authors as follows:

Response: We greatly appreciate the reviewer's insightful comments and advice aimed at improving our manuscript. In response to the reviewer's feedback during the second round of revision, we have made modifications accordingly (removing the inappropriate sentence explained below). We are truly grateful for the reviewer's detailed feedback on our responses, their encouraging words, and the time they spent further clarifying previous comments and suggestions.

To 1: I acknowledge the efforts of the authors and the interesting insights from following my suggestion. I think the presentation of the results as additional supplementary figure adds an important value on the physiological transcriptional regulation of TM4SF19.

To 2: The authors now present a complete picture on the regulation of macrophage Trem2 and Lyve on gene expression and protein level during co-culture with adipocytes. I acknowledge that. In addition, I understand the technical limitations the authors encountered during the isolation of the distinct cell populations for ex vivo phagocytosis assay and I acknowledge their attempts. With regards to my concerns of choosing C1qa, C1qb and C1qc as phagocytosis markers, I highly appreciate that the authors meanwhile considered Mertk, Cd163 as phagocytosis markers and updated the corresponding figures. In light with the additional point raised in the discussion, I think my concern is well addressed.

To 3: Adding the control group to Revised Figure 5J and supplemental Video 2 was essential in order to keep the conclusion that clearance of dying/dead adipocytes by macrophages is increased in TM4SF19 KO mice. I acknowledge the additional work of the authors.

To 4: I apologize to the authors for missing the data on mitochondrial metabolism in the previous version of the manuscript (Sup Fig 9.) I acknowledge, that the authors include these data in the new Figure 7, clearly supporting their conclusion. However, as the sentence in line 316-320 "These

data indicate that efficient efferocytosis of Tm4sf19 KO macrophages mitigates the proinflammatory environment induced by HFD, promotes tissue expansion by adipocyte hyperplasia, and restores mitochondrial metabolism and insulin sensitivity within the adipocytes of obese adipose tissue." Still mentions the restoration of mitochondrial metabolism without any reference to Figure 7, I suggest to delete this part of the conclusion at this position in the manuscript and rather add it after the presentation and description of Figure 7. On top of that, the same is true for insulin sensitivity. Also here, the conclusion is drawn before the actual data has been presented. In line with my previous suggestion, I again propose that the authors delete this part of the conclusion here.

Response: We agree with the reviewer's critique and opinion; thus, the sentence has been removed from the manuscript.

To 5: I appreciate the clarification of this point and agree, that cell-autonomous effects do not significantly contribute to the phenotype of adipocytes in TM4SF19 KO mice and that rather potentially increased adipogenesis might be causal to the observed phenotype. Still, with regards to the TM4SF19 KO – mediated potential increase of de novo adipogenesis from progenitors, can the authors speculate, how this effect would be mediated. Along that line, perhaps it would be interesting to follow adipogenic differentiation of SVF fraction in the presence or absence of WT vs. vs TM4SF19 KO macrophages or their conditioned medium.

Response: Thank you for your insightful suggestion. We intend to explore the impact of TM4SF19 knockout in macrophages on the adipogenic differentiation of progenitors in a follow-up study. Additionally, we anticipate the possibility of identifying secretory factors that promote adipogenesis through the analysis of conditioned media from Tm4sf19 knockout macrophages.

In line with this hypothesis, our recent work in brown adipose tissue (Rayanne B Burl, et.al, **Deconstructing cold-induced brown adipocyte neogenesis in mice** *eLife* 11:e80167) indicates that adipogenic niches are created by the interaction of adipocyte progenitors with recruited lipid handling macrophages and dendritic cells, which drive progenitor proliferation at sites of adipocyte efferocytosis. Interestingly, spatial transcriptomic analysis indicates that the departure of immune cells from the niche is closely associated with the rapid differentiation of newly-divided progenitors. Thus, we speculate that the rapid efferocytosis observed in TM4sf19 KO mice would accelerate the de novo adipogenesis under conditions of HFD feeding.

To 6: I thank the author for highlighting that the observed body weight differencers might be the result of a complex feedback relationship that begins with Tm4sf19 function in macrophages.

To 7: Thanks for adding the additional data and clearing this point.

To 8: I highly appreciate the effort of the authors to reanalyze data on energy expenditure using

regression based analysis of covariance (ANCOVA) with body mass as a covariate as described in PMID: 22205519 as well as their efforts to investigate potential differences in intestinal lipid uptake and have no further concerns.

To 9: Deletion of Trem2+ indeed avoids conclusion.

To 10: I understand that this might go beyond the scope of this manuscript. However, if I am not mistaken, in the revised manuscript, the authors also present data generated from livers of WT and TM4SF19 KO- mice showing that the inflammatory response of HFD is indeed reduced in the KO similarly as observed in the WAT.

Response: The reviewer correctly pointed out that our original manuscript included some of the results on the liver phenotype of Tm4SF19 knockout as an obesity-related phenotype. Previously, we had assumed that the reviewer was inquiring about phenotypes beyond those related to adipose tissue and liver. Moving forward, we will persist in investigating the effects of TM4SF19 knockout and overexpression on other metabolic diseases and within the immunological context.

To Minor Fig2c: I appreciate the efforts of the authors.

To Minor FigS2: The point was addressed and I do not have any further concerns.

Thanks for giving more technical details on the phagocytosis assays. I understand the rational of the experiments and do not have any further concerns.

I acknowledge that all other minor points were sufficiently addressed.

Reviewer #2 (Remarks to the Author):

The author has addressed the issues and the manuscript is ready for publication.

Response: We deeply appreciate the insightful comments and advice provided by the reviewer, which have greatly contributed to the improvement of our manuscript.

Reviewer #3 (Remarks to the Author):

The authors have made substantial efforts, enhancing the clarity and comprehensibility of the manuscript significantly. This revision by Choi et al. has successfully addressed all the concerns I previously raised. The paper now stands out in terms of both quality and importance. Given its notable improvements and the relevance of its findings, I recommend its acceptance for publication in Nature Communications.

Response: We deeply appreciate the insightful comments and advice provided by the reviewer, which have greatly improved our manuscript. We are truly grateful for the reviewer's encouraging

words and considerations.